# Prevalence, lived experiences and user profiles in e-cigarette use: A mixed methods study among French college students

Shérazade Kinouani[1,2]*, Héléna Da Cruz[1], Emmanuel Langlois[3], Christophe Tzourio[1]

1 University of Bordeaux, Inserm, Bordeaux Population Health Research Center, Team HEALTHY, UMR 1219, Bordeaux, France, 2 Department of General Practice, University of Bordeaux, Bordeaux, France, 3 University of Bordeaux, CNRS, Emile Durkheim Center, UMR 5116, Bordeaux, France

* sherazade.kinouani@u-bordeaux.fr

## Abstract

### Background

Little is known about e-cigarette use in French students. Our aims were to estimate the prevalence of e-cigarette experimentation and current e-cigarette use; describe the reasons for using e-cigarettes; explore the vaping experience and identify the profiles of e-cigarette users.

### Methods

We used a sequential, explanatory mixed methods design in a sample of French college students. Quantitative data was collected online for a cross-sectional analysis among 1698 students. Two separate analysis based on the thematic analysis and the Grounded Theory were also performed in 20 semi-structured interviews, focusing former and current smokers also current vapers.

### Results

The prevalence of e-cigarette experimentation was 39.3% (95% CI: 35.2–44.0) and 5.1% (95% CI: 3.2–8.0) of students were current e-cigarette users. Experimentation was opportunistic while current usage was rational, requiring to acquire a personal electronic device, getting used to its technicality, appreciating its availability, discretion, and learning the practice. In this context, three distinct groups of e-cigarette users were identified, based on assumed identity, tobacco and e-cigarette use, the functions assigned to e-cigarettes, and intentions with regards to vaping in the future.

### Conclusion

Despite some limitations mainly related to the participants self-selection, this research showed that while many smokers and former smokers have tried e-cigarettes in this student population, few have continued to use them continuously. Moreover, these current e-cigarette users were a heterogeneous group. Longitudinal studies are needed in young adult

**Data Availability Statement:** Data cannot be shared freely as the authors have to comply with French regulations (the Commission Informatique et Libertés, which is the French authority that

ensures that the collection of data in the context of research does not infringe on the freedoms, rights, and privacy of individuals/number: DR-2013-019) and European regulations (General Data Protection Regulation). Thus, the qualitative component (interviews) cannot be shared but data of the quantitative component of the study are available upon reasonable request. The i-Share project data used in the quantitative component can be requested for scientific collaboration, following a procedure described on the study website (https://research.i-share.fr/how-to-collaborate/). All requests must be sent to the i-Share Scientific Collaborations Coordinator: Ilaria Montagni (ilaria.montagni@u-bordeaux.fr).

**Funding:** This work was supported by the French National Cancer Institute [grant number INCa_11502]. The i-Share team is currently supported by an unrestricted grant of the Nouvelle-Aquitaine Regional Council (Conseil Régional Nouvelle-Aquitaine) [grant number: 4370420] and by the Bordeaux "Initiatives d'excellence" (IdEx) program of the University of Bordeaux [ANR-10-IDEX-03-02]. It has also received grants from the Nouvelle-Aquitaine Regional Health Agency (Agence Régionale de Santé Nouvelle-Aquitaine) and Public Health France (Santé Publique France). The funding bodies had no role in study design, data collection, analysis and interpretation, decision to publish or preparation of the manuscript.

**Competing interests:** The authors have declared that no competing interests exist.

smokers for a better understanding of how their tobacco and e-cigarette use affect each other and change over time.

## Introduction

E-cigarette use (or vaping) has increased worldwide, particularly among young adults [1,2]. Although toxicological studies suggest it is less harmful than smoking, the risks of long-term on human health are as yet unknown [3,4]. Several observational studies have suggested that vaping could be associated with a later cigarette smoking in never smokers [5,6] or a later risk of relapse in former smokers [7,8]. It is probable that the risk-benefit balance of chronic vaping will always be disadvantageous in never smokers who vape, as well as in current smokers who do not quit smoking completely, engaging in dual use of tobacco and vaping [9,10]. Despite these uncertainties, e-cigarettes are reported to be the most popular tool among young Europeans to quit smoking. Among current tobacco smokers who participated to the Eurobarometer Survey in 2020, those aged 15–24 were less likely to have attempted to stop smoking compared with those aged ≥ 25 years. While nearly three quarters of Europeans ≥15 years old who tried to quit smoking did so without any help, 29% used a cessation aid. The most frequently used aids were: first pharmacotherapy and second e-cigarettes. Those aged 15–24 were less likely to use pharmacotherapy and more likely to use e-cigarettes than those aged ≥55 years [11]. These results are roughly comparable to what was described by the same survey in 2017 [12].

According to the French national public health agency *Santé Publique France*: 37% of French people aged 18–75 tried e-cigarettes in 2020; 5.4% were currently using them, of which three quarters (4.3%) were using them daily [13]. E-cigarettes are not considered as medical devices in France but consumer products whose use, sale and advertising remain strongly regulated. Vaping products (e-liquids and electronic devices) on the French market comply with European regulations since May 2016 setting maximum nicotine content in e-liquids at 20 mg/ml [14]. Vaping is authorized everywhere, except in places frequented by minors, closed spaces or places where use is prohibited by the internal rules of the establishment. Selling to people under 18 is also prohibited, even online and advertising for vaping products has been banned. Compared to other countries, the French regulatory context can be seen as "moderate" regarding e-cigarettes. Unlike the United Kingdom, neither the legislative framework nor the health agencies promote e-cigarettes as a smoking cessation tool. However, they are not considered tobacco products, as in Mexico or Turkey. The French position on the regulation of e-cigarette use should be interpreted in the light of the evolution of tobacco use in recent years. In 2020, the prevalence of daily tobacco smoking was 25% among French adults [13]. Although this prevalence has decreased over the last 20 years, it remains among the highest in Europe [15].

Our study aimed to understand how e-cigarettes were used and perceived by French college students in their particular context. First, we wanted to estimate the levels of experimentation and current use of e-cigarettes in this population, which were not known. The prevalence were estimated in the whole sample and then according to smoking status. Second, we wanted to interpret these prevalence by taking into account initial intentions to vaping, the evolution of these intentions over time, and the experience of students who have maintained continuous use of e-cigarettes over a few months. Our general aim was to describe e-cigarette use in a French student population. The specific objectives were: i) estimating prevalence of e-cigarette experimentation and current e-cigarette use; ii) describing reasons for using e-cigarettes; iii)

exploring vaping experiences and identifying e-cigarette user profiles in current and former smokers.

## Materials and methods

We conducted a sequential explanatory mixed methods study: QUANT → qual [16]. This article follows STROBE recommendations for reporting observational studies [17] and COREQ for reporting qualitative research [18].

### Data sources and participants in the quantitative phase

We carried out an online quantitative study among students who had already participated in the i-Share research project (Internet-based Students Health Research Enterprise), an ongoing e-cohort of French speaking students (French speaking students with French nationality like French speaking international students): www.i-share.fr. All students included in the i-Share project between February 2013 and January 2016 were contacted by e-mail to participate in our ancillary study on vaping. To be included in our analysis, the volunteers had to be at least 18, know how to read and understand French, and declare themselves enrolled in a higher education institution from one French university (University of Bordeaux). Data were collected between February and April 2016.

### Data collection and participants in the qualitative phase

Students at University of Bordeaux were invited by an advertisement via university social networks or e-mails. Those who answered the quantitative phase were also contacted by e-mail if they had reported trying e-cigarettes. Finally, we asked interviewed students if they knew other users who might be interested in participating (snowball strategy). To be included, volunteers had to be current e-cigarette users or have used them frequently for at least two continuous months, be studying at Bordeaux University, and be current or former smokers. We obtained a purposive sample based on three criteria: gender, field of study and smoking status. Smoking status included recent former smokers (had quit less than one year before), older former smokers (had quit for at least one year or more), occasional smokers (smoked less than one cigarette per day) and daily smokers.

Semi-structured individual interviews took place between April 2016 and June 2017. Two trained medical students involved in the research team led all interviews because of their proximity in age to the respondents. Interviews were audio-recorded and then transcribed. The initial guide was drafted by the research team based on the literature and the first results of the quantitative phase. It was then modified as interviews were conducted, and new hypotheses emerged. The final version is available as supporting information (S1 and S2 Files). At the end of the eighteenth interview, we seemed to have reached data saturation. Two other interviews were added, without any new themes or categories emerging.

### Analyses

**Quantitative component.** The main outcomes were experimentation with e-cigarettes (defined as trying at least once in a lifetime) and current use of e-cigarettes (defined as daily or occasional use of e-cigarettes). We analyzed several sociodemographic, economic, academic, and medical characteristics from the i-Share baseline questionnaire when they were available (S3 and S4 Files). We also analyzed ancillary study data on smoking status and e-cigarette use (age of first try, reasons for trying e-cigarettes, current use). In current e-cigarette users, we explored reasons for using e-cigarettes, use frequency, nicotine use in e-liquids, vaping places

and times. We described continuous variables using median and interquartile range (IQR) and categorical variables using numbers and proportions. Bivariate comparison was performed by chi-square test. First, we described the sociodemographic, economic, academic, and medical characteristics in the whole sample. Secondly, we estimated the prevalence of e-cigarette experimentation and current e-cigarette use with 95% confidence interval (95% CI). We calculated them before and after weighting by calibration on the known margins of the student population at the University for the 2015–2016 academic year [19]. This calibration was carried out with a program developed by the French National Institute for Statistics and Economic Studies designed to take into account the non-response bias: the macroSAS CALMAR® [20]. Calibration variables were gender, age, and study fields. Thirdly, we described e-cigarette use in those students who have experimented with it and associations between e-cigarette use and smoking status were analyzed in this subsample. All p-values were two-tailed and we considered $p < 0.05$ to be statistically significant. All statistical analyses were performed with R® (version 4.0.2).

**Qualitative component.** A pseudonym was assigned to each participant during interview transcription and used when attributing quotes. We carried out two separate but concomitant analyses of the same data. First, we performed a thematic analysis on reasons for using e-cigarettes, starting at the end of data collection of the quantitative phase [21]. An analysis based on Grounded Theory was also performed to understand the lived experiences and user profiles [22,23]. Thematic analysis is an analytical method allowing to both test the motives for using e-cigarettes identified in the quantitative phase and to highlight convergences and divergences between the motives for experimenting and those for continuing to use electronic cigarettes. On the other hand, the analysis inspired by Grounded Theory seemed more appropriate to bring out the conceptualizing categories describing the lived experiences of vapers or their user's profile. Analyses were carried out by five trained researchers, either manually or using Nvivo 10 ®. Each interview was coded individually by at least two of the five researchers with iterative pooling times, until data saturation. The themes or categories obtained as the analyses progressed and final theorizing about user profiles were discussed by all co-authors. A synthesis of the study, its results and their interpretation were e-mailed to all interviewed students in April 2018 for comments.

**Integrative mixed methods analysis.** We adopted a building approach to the results [16], beginning by collecting and analyzing quantitative data and then using these findings to guide data collection and analysis in the qualitative phase. The quantitative phase described the prevalence and reasons for using e-cigarettes in our student population. Reasons identified in the quantitative component were investigated during the thematic analysis of data from the qualitative component. An analysis based on Grounded Theory focused in parallel on lived experiences of vaping among university students who regularly used e-cigarettes (for at least two continuous months), whether they were dual users, former smokers and relapsing smokers.

## Ethics

This research was conducted in accordance with the Declaration of Helsinki. All subjects in the quantitative phase gave their informed consent before participating in the i-Share project and ancillary studies. The i-Share project protocol was approved by the *Commission Informatique et Libertés*, the national authority that ensures that data collection in research does not violate freedoms, rights, and human privacy (number: DR-2013-019). Students participating in the online quantitative study on e-cigarettes received points that can be exchanged for cinema tickets or fruit and vegetable hampers. The qualitative phase received ethical approval from the ethics committee of Bordeaux University Hospital, France (number: GP-CE 2018/17). All subjects gave their oral consent at the beginning of the audio-recording.

## Results

### Quantitative phase

The quantitative component comprised 5214 students invited to participate in the ancillary study on e-cigarette use; 1815 subjects answered (response rate: 34.8%) and 1698 students were finally kept for the analyses (Fig 1).

More than 3/4 of these students were female. More than two out of five students were freshmen and 90% were 24 or under (median age: 21; IQR: 19.0–23.0). Just under half (46%) were in the healthcare field and the parents of 53% of them had had higher education. More than 80% of them perceived their health as good or very good. Table 1 summarizes their characteristics.

The weighted prevalence of e-cigarette experimentation in the student population was 39.3%, 95% CI: 35.2–44.0 (Table 2). The median age of first trying e-cigarettes was 20, IQR: 18.0–21.0. There was more experimentation in former and current smokers than in never smokers (Table 3). Curiosity, the opportunity to try and the diversity of flavors were the main reasons for experimenting with them (S1 Table).

The weighted prevalence of current use in the student population was 5.1%, (95% CI: 3.2–8.0) (Table 2). Current use was most frequent in former smokers, followed by current smokers,

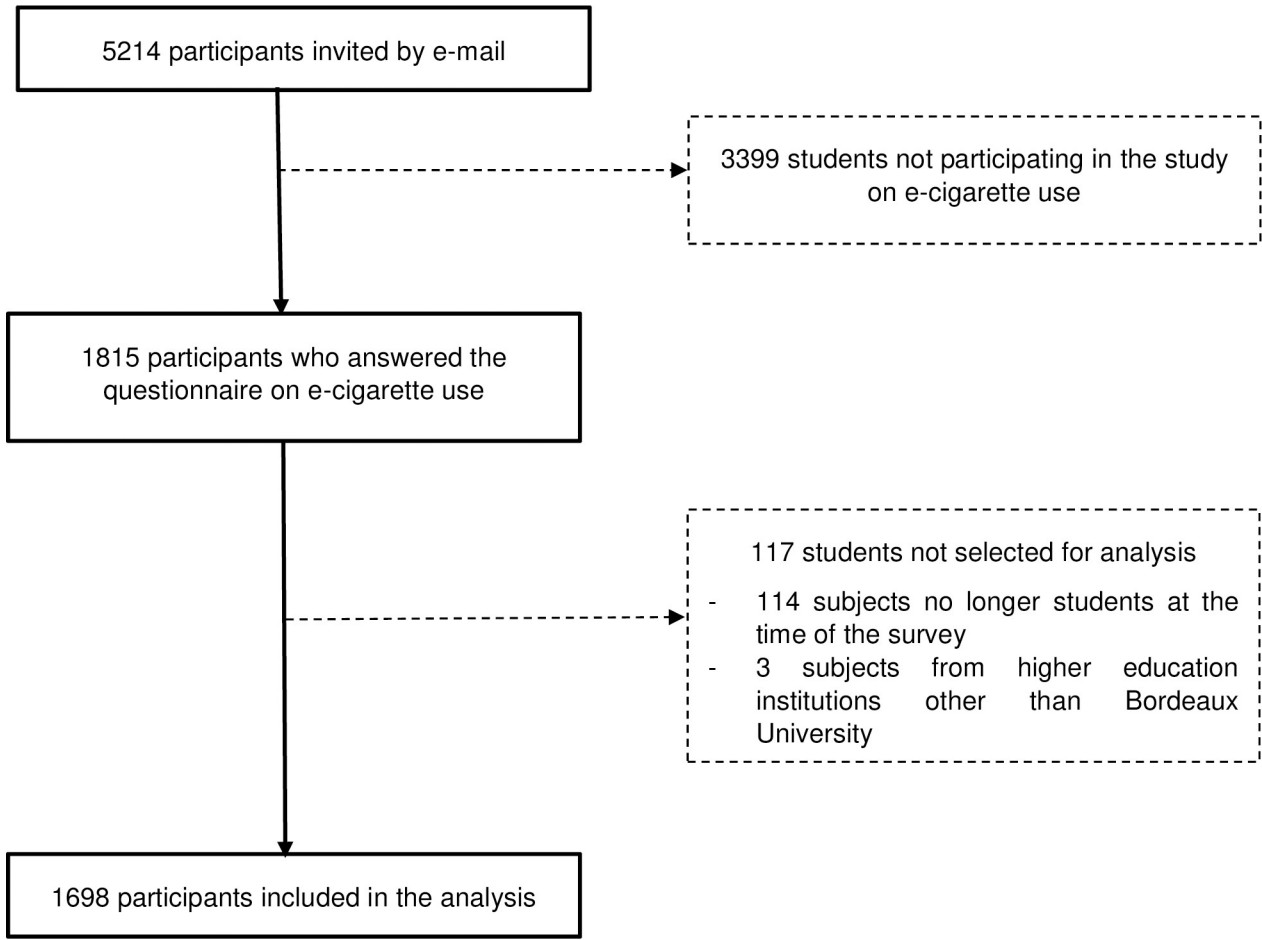

**Fig 1. i-Share project students participating in the ancillary quantitative study on e-cigarette use at the University of Bordeaux (France) in 2016.**

**Table 1. Characteristics of college students who participated in the quantitative study on e-cigarette use at the University of Bordeaux in 2016, N = 1698.**

| Characteristics | n | % |
|---|---|---|
| Duration between the inclusion in i-Share project and those in the ancillary study, in weeks (median, IQR*) | 60 | 25–102 |
| Gender [α] | | |
| • Men | 375 | 22.1 |
| • Women | 1323 | 77.9 |
| Age, in years [α] | | |
| • 18–20 | 775 | 45.7 |
| • 21–24 | 768 | 45.2 |
| • 25 and over | 155 | 9.1 |
| Academic study fields [α] | | |
| • Healthcare | 774 | 45.6 |
| • Literature arts, Humanities and social sciences | 391 | 23.0 |
| • Sciences | 214 | 12.6 |
| • Economics, management and law | 133 | 7.8 |
| • Other | 186 | 11.0 |
| Academic year of study [α] | | |
| • 1st year | 754 | 44.4 |
| • 2nd year | 340 | 20.0 |
| • 3rd year | 234 | 13.8 |
| • Beyond 3rd year | 342 | 20.2 |
| • Other | 28 | 1.6 |
| Students' living conditions [α] | | |
| • In apartment: couple, or colocation | 462 | 27.2 |
| • In apartment, alone | 576 | 33.9 |
| • Parents' home | 409 | 24.1 |
| • University residence | 183 | 10.8 |
| • Other | 68 | 4.0 |
| Student economic resources, in multiple choice [α] | | |
| • Family | 1386 | 81.6 |
| • Scholarship | 753 | 44.3 |
| • Paid job (including summer job, paid internship) | 677 | 39.9 |
| • Other | 103 | 6.1 |
| Parents' educational level [α] | | |
| • Higher education, university | 897 | 52.8 |
| • High school or vocational study | 762 | 44.9 |
| • Primary education | 10 | 0.6 |
| • I don't know | 29 | 1.7 |
| Self-rated of current health [α] | | |
| • Very good to good | 1380 | 81.3 |
| • Fair | 275 | 16.2 |
| • Poor to very poor | 43 | 2.5 |
| Self-rated of sleep quality over the past 3 months [α] | | |
| • Good | 953 | 56.1 |
| • Neither good nor poor | 420 | 24.7 |
| • Poor | 325 | 19.2 |
| Diagnosis previously made by a physician [α] | | |
| • Headaches | 355 | 20.9 |

*(Continued)*

**Table 1.** (Continued)

| Characteristics | n | % |
|---|---|---|
| • Asthma | 334 | 19.7 |
| • Anxiety and phobic disorders | 236 | 13.9 |
| • Depression | 174 | 10.2 |
| Smoking status [β] | | |
| • Current smokers | 833 | 49.3 |
| • Former smokers | 164 | 9.7 |
| • Never smokers | 694 | 41.0 |
| Current alcohol use frequency [α] | | |
| • Never | 100 | 5.8 |
| • ≤ Monthly | 429 | 25.3 |
| • > Monthly but not weekly | 842 | 49.2 |
| • > Weekly but not daily | 319 | 18.8 |
| • Daily | 8 | 0.5 |
| Cannabis use at least once in lifetime [α] | 884 | 52.1 |

[*] *IQR: Interquartile range;*

[α]*Data collected at baseline in the i-Share project between February 2013 and January 2016;*

[β]*Data collected in the ancillary quantitative study on e-cigarette use.*

but was rare in never smokers (Table 3). The majority of the 58 current users in the quantitative study vaped e-liquids containing nicotine, but five did not know if their e-liquids contained nicotine (S1 Table). There rarely seemed to be a single reason to continue using e-cigarettes. The four predominant reasons for current use reported by students were either related to their ease of use in time and space or related to the management of addiction symptoms (S1 Table).

## Qualitative phase

**Sample.** As shown in Table 4, 20 students were interviewed in the qualitative component, 11 men and 9 women (median age: 26; IQR: 23.7–28.0). The interviews lasted on average 55.25 minutes. Fourteen students were former smokers and six were dual users, combining vaping with smoking. One of these dual users was a relapsed tobacco user.

**Table 2. Prevalence of e-cigarette use among college students at the University of Bordeaux in 2016, N = 1694[*].**

| E-cigarette use | n | Crude % | 95% CI [β] | Weighted % [α] | 95% CI [β] |
|---|---|---|---|---|---|
| No experiment [σ] | 991 | 58.5 | 56.1–60.9 | 55.6 | 51.3–60.0 |
| Experiment [γ] | 645 | 38.1 | 35.8–40.4 | 39.3 | 35.2–44.0 |
| Current use [ω] | 58 | 3.4 | 2.6–4.4 | 5.1 | 3.2–8.0 |
| • Occasional use | 28 | 1.6 | 1.1–2.4 | 2.4 | 1.2–5.0 |
| • Daily use | 30 | 1.6 | 1.2–2.5 | 2.7 | 1.5–5.0 |

[*]*1698 subjects included in analysis but only data concerning 1694 subjects were available on e-cigarette use and calibration variables.*

[α] *Weighting by calibration on margins, using the MacroSAS Calmar® program (raking ratio method). The calibration variables were: Gender, age and the study fields;*

[β]*95% confidence interval;*

[σ] *Never tried to use e-cigarettes;*

[γ] *Having tried at least once to use e-cigarettes;*

[ω] *Occasional (<1 time per day) or daily (≥ 1 time per day) use of e-cigarettes.*

## Thematic analysis in the qualitative phase

Overall, three main themes were identified as reasons for experimenting with e-cigarettes (Fig 2): reasons related to their features; reasons related to nicotine delivery or tobacco use; and reasons related to the convenience of vaping and social interactions. The opportunity to try and reasons related to the features of e-cigarettes had a major influence on the first try, even in those who finally quit smoking (S1 Table). Moreover, vaping had seemed less expensive than smoking or easier and more pleasant than pharmacotherapy (S1 Table). Friends, family or a partner who already used e-cigarettes greatly contributed to their initiation.

The same three main themes were found as reasons for pursuing e-cigarette use (Fig 3). However, their description was richer than for the experiment: the reported reasons were here multiple for each student. Whether dual users or former smokers, e-cigarette current use was explained mainly by comparing to tobacco (Fig 3, S1 Table). It was a way for some students to maintain their smoking habits, even when no longer smoking tobacco. Some said it helped break the day up. Others said it allowed them to continue enjoying the same gestures and sensations. It also produces the psychotropic effects of nicotine and helps users focus while working, but also relax. Many downsides of smoking were circumvented by e-cigarette use. Some saw e-cigarettes as less troublesome for those around them, owing to the lack of smell, or as a way around the ban on smoking in public places. Because they provide nicotine, e-cigarettes were also seen as a good substitute allowing a gradual reduction in tobacco or nicotine use, sometimes until cessation. Other reasons were unrelated to tobacco use, such as the possibility to customize use or the fact that vaping becomes a pleasure or leisure behavior (Fig 3).

## Grounded theory method applied in the qualitative phase: Vaping but not necessarily being a vaper

*Becoming a persevering e-cigarette user.* Three categories emerged: investment in a personal electronic device; seeking information; electronic device properties.

**Table 3. Prevalence of e-cigarette use according to smoking status among college students at the University of Bordeaux in 2016, N = 1691*.**

| E-cigarette use | n | Crude % | p-value [β] | Weighted % [α] | p-value [β] |
|---|---|---|---|---|---|
| No experiment [σ] | 989 | 58.5 | | 55.6 | |
| • Never smokers | 593 | 85.4 | <0.0001 | 82.4 | <0.0001 |
| • Former smokers | 47 | 28.7 | | 29.9 | |
| • Current smokers | 349 | 41.9 | | 40.7 | |
| Experiment [γ] | 644 | 38.1 | | 39.3 | |
| • Never smokers | 99 | 14.3 | <0.0001 | 17.4 | <0.0001 |
| • Former smokers | 97 | 59.1 | | 55.5 | |
| • Current smokers | 448 | 53.8 | | 52.6 | |
| Current use [ω] | 58 | 3.4 | | 5.1 | |
| • Never smokers | 2 | 0.3 | <0.0001 | 0.2 | <0.0001 |
| • Former smokers | 20 | 12.2 | | 14.6 | |
| • Current smokers | 36 | 4.3 | | 6.7 | |

*1698 subjects included in analysis but only data concerning 1691 subjects were available on e-cigarette use, smoking status and calibration variables.

[α] *Weighting by calibration on margins, using the MacroSAS Calmar® program (raking ratio method). The calibration variables were: Gender, age and the study fields;*

[β] *Chi square test;*

[σ] *Never tried to use e-cigarettes;*

[γ] *Having tried at least once to use e-cigarettes;*

[ω] *Occasional (<1 time per day) or daily (≥ 1 time per day) use of e-cigarettes.*

**Table 4. Characteristics of college students participating in the qualitative study on e-cigarette use at the University of Bordeaux in 2016–2017, N = 20.**

| Characteristics | n (%) |
|---|---|
| Age, in years | |
| • 19–25 | 10 (50) |
| • 26–29 | 10 (50) |
| Gender | |
| • Men | 11 (55) |
| • Women | 9 (45) |
| Participants in i-Share project [α] | 7 (35) |
| Academic study fields | |
| • Healthcare | 11 |
| | *Medicine*: 9 |
| | *Public health*: 1 |
| | *Pharmacy*: 1 |
| • Humanities and social sciences | 2 |
| • Sciences | 3 |
| | *Life sciences*: 2 |
| | *Information technology*: 1 |
| • Economics, management and law | 2 |
| • Literature and arts | 2 |
| Smoking status | |
| • Former smokers, > 1 year | 8 (40) |
| • Former smokers, ≤ 1 year | 6 (30) |
| • Daily smokers (≥ 1 cigarette per day) | 3 (15) |
| • Occasional smokers (< 1 cigarette per day) | 3 (15) |
| E-cigarette use | |
| • Former users | 4 (20) |
| • Daily users (≥ 1 inhalation per day) | 14 (70) |
| • Occasional users (<1 inhalation per day) | 2 (10) |

[α] *Seven college students participating in i-Share project answered the ancillary quantitative study about e-cigarette use and also accepted to be interviewed in the qualitative phase.*

Students sometimes used a friend's or relative's device during experimentation but had to have their own device for regular use. They made their first purchase in stores or on Internet; their first personal e-cigarette was rarely a gift. Whatever its origin, becoming current use meant having its own device. Over time, some students changed devices and opted for technically more efficient models, offering better rendering of flavors, aerosol density, throat hit or battery life.

Current e-cigarette users often showed little interest in knowing more about vaping. When they did inquire, they mainly wanted information about the health effects, variety of flavors and ways to improve device performances or reduce the cost of use. The lack of interest in staying informed was claimed by some students as a way to keep a distance from their own usage and other users (the vaping community).

Three properties mainly characterized their electronic devices: discretion, availability, and above all, technicality. Discretion is ensured by less smoke production, no smell on clothing, or the small size of devices. It is easy to go to a store for devices, spare parts or e-liquid refills, or to buy them online.

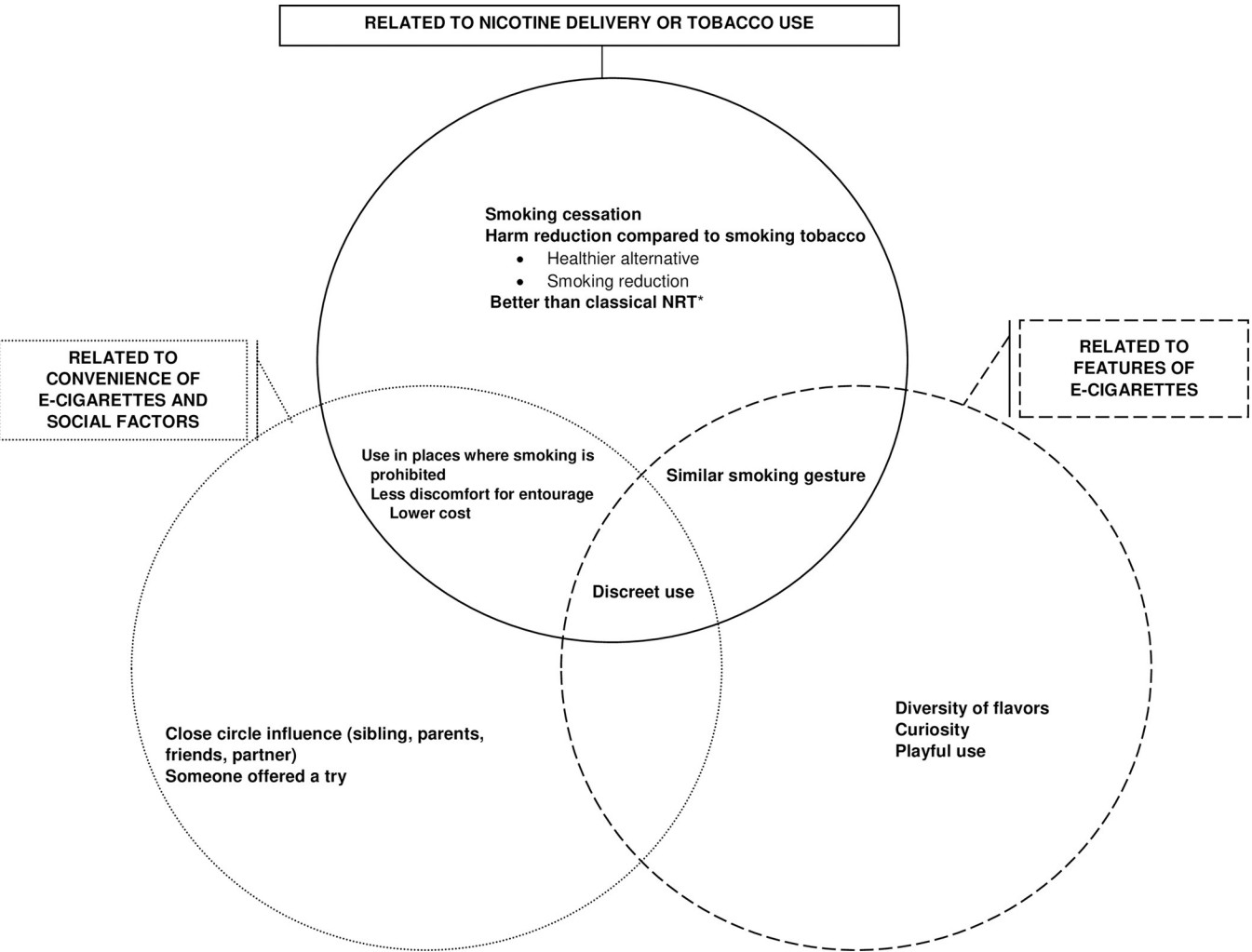

**Fig 2. Reasons for trying e-cigarette use among current and former tobacco smokers of the University of Bordeaux, N = 20.** *NRT: Nicotine replacement therapy.

"*I find that either ordering on Internet or going to a real store where you can try several tastes and talk to the seller. . . well, it's more in the spirit of electronic cigarettes than going to buy an electronic cigarette and e-liquid from the tobacconist* " (Tao: man, former smoker and e-cigarette user for 4 months).

The technicality of devices was both a strength and weakness. Users needed to learn how to inhale the aerosol correctly for the expected effects, as with tobacco cigarettes, and also how to set and maintain their electronic device. This generated recurring malfunctions, repeated maintenance, and a personal effort to acquire technical skills, which discouraged some. Levels of personal interest in knowing more about vaping and electronic devices resulted in distinct attitudes. Either partial or total smoking substitution by vaping, conserving smoking habits (frequency, times and places) as much as possible; or adopting vaping as a new behavior, different from smoking and perceived as enjoyable.

*Self-image and social interactions of current users*. Vaping allowed some former smokers to manage their tobacco addiction. They had succeeded in replacing cigarettes with e-cigarettes

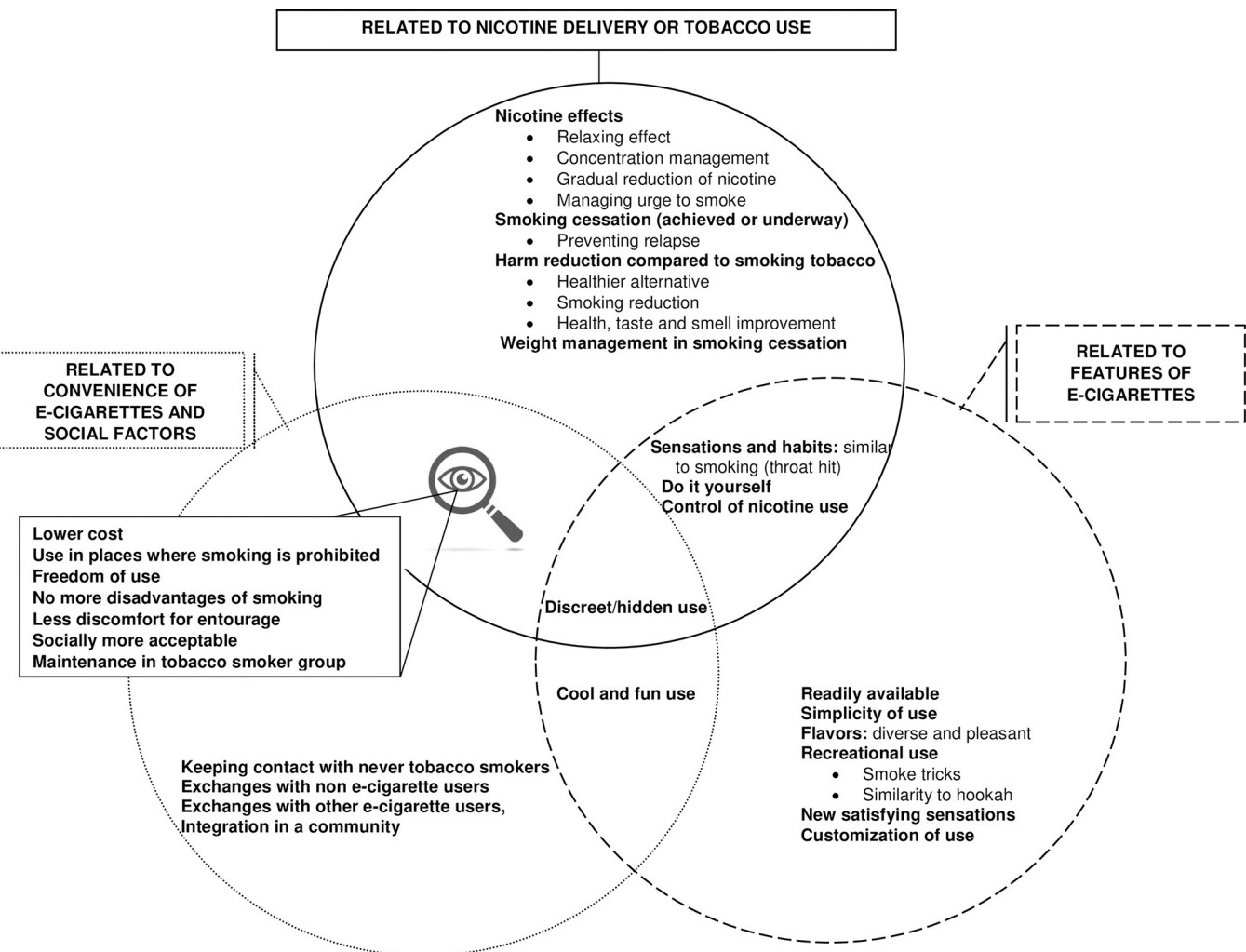

**Fig 3. Reasons for pursuing e-cigarette use among current and former tobacco smokers of the University of Bordeaux, N = 20.**

and felt in control again. Other former smokers saw a lack of freedom, however, as e-cigarettes fed their addiction.

> "*I think in a way yes, they (e-cigarettes) allowed me this first stop, and that made me say: well you can stop that (tobacco)"* (Anna: woman, former smoker who also quit e-cigarettes).

> "*Well, you don't stress out about your pack of cigarettes anymore but you stress out. . . if your thing (her e-cigarette) is loaded. Well, it's. . . it's the same as it is. . . we're just so addicted to something all the time. And then it remains the same substance: nicotine* " (Bea: woman, former smoker and e-cigarette user for 2 years).

Users felt that vaping drew attention to themselves, and this was variably received. While some saw vaping as a way of giving a trendy image of themselves, others thought it made them look ridiculous, weak (vaping instead of smoking a real cigarette) or naive (vaping a dangerous product, regardless of health risk).

Non-users often showed surprise and curiosity during social interactions. They were tolerant and even wanted to know more about e-cigarette use: vaping was perceived overall as less

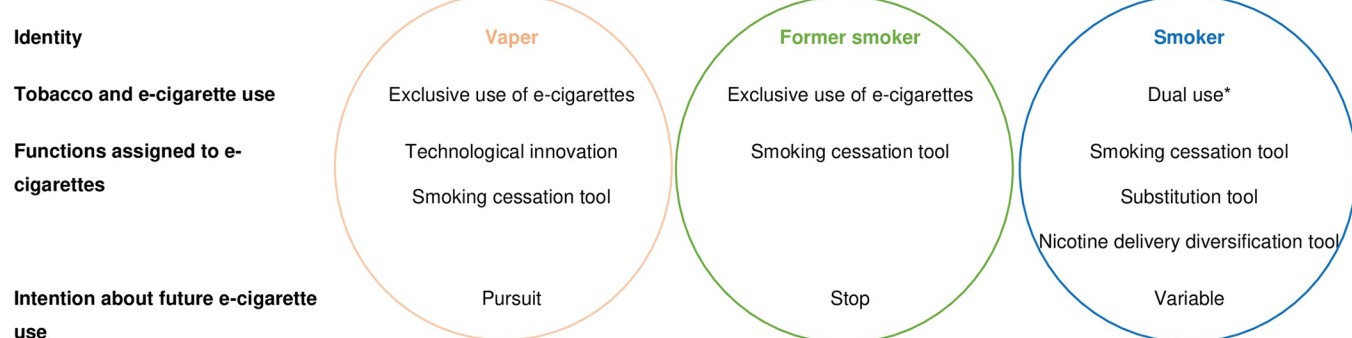

**Fig 4. Relationship between user identity, the function assigned to e-cigarettes and the change in tobacco and e-cigarette use, among current and former smokers of the University of Bordeaux N = 20.** * Concomitant use of tobacco and e-cigarettes.

harmful than smoking. Vaping favored social interactions between e-cigarette users. They shared knowledge, helped each other solve technical problems, talked about experiences of quitting smoking, etc. Some comments suggested membership of a subculture, notably via use of specific technical vocabulary to describe their device and practice, in particular among former smokers with a very personalized use of e-cigarettes. They were also those who described themselves as "vapers".

*Current user profiles*. Three groups of students were distinguished, based on the four categories emerging from the analysis (Fig 4): function assigned to e-cigarettes; concomitant use of tobacco and e-cigarettes; intention about future e-cigarette use; identity.

Firstly, some users had succeeded in stopping smoking through e-cigarettes and saw them primarily as a technological innovation. Accumulating vaping knowledge and technical skills contributed to constructing their vaper identity. Vaping was a whole new experience, both personally and socially. They knowingly continued to use e-cigarettes after quitting smoking. Secondly, other former smokers rejected the vaper label. E-cigarettes were only a tool to stop smoking. They planned to quit smoking and achieved it through e-cigarettes or nicotine replacement therapies (if vaping was unsuccessful). These former smokers did not want vaping to participate in the construction of their identity, seeing vaping as a temporary stage. Many stopped using e-cigarettes after quitting smoking. Others continued to use e-cigarettes mainly for fear of relapse. Unlike the first two groups, common characteristics of the third group were their smoker identity and dual use. Based on the function assigned to e-cigarettes, three kinds of dual users were observed. Some perceived e-cigarettes as a tool to stop smoking and hoped that dual use was a step towards smoking cessation. Other dual users replaced tobacco by vaping except in circumstances where they felt that smoking could not be replaced, such as stressful events or evenings. The last dual users were not at all quitting or switching. E-cigarettes were an additional way of diversifying their nicotine use, particularly because of its practicality, the possibility of testing flavors, and so on.

## Integrative phase: From opportunistic experimentation to rational current use

According to the quantitative phase, experimentation with e-cigarettes was common among student former and current smokers. Although lesser, it also existed among never smokers. Quantitative and qualitative phases both suggested that experimentation with e-cigarettes was opportunistic. It was mainly favored by the student curiosity, a close circle who already used e-cigarettes, the diversity of flavors, or the playful aspect of practice. Only one in 20 students

reported current e-cigarette use in the quantitative phase. According to the qualitative phase, this current use required: acquiring one's own electronic device, getting used to its technicality, appreciating its availability, discretion, and learning the practice. Such personal investment explained why current use was marginal among student never smokers in our quantitative phase. To persevere in vaping, the experimenter indeed had to find enough advantages in use of e-cigarettes. After comparing vaping to smoking, former and current smokers all found many reasons to continue using them. Finally, vaping did not necessarily mean considering yourself to be a vaper. Students chose to pursue long-term e-cigarette use based on their assumed identity of smoker, former smoker, or vaper.

## Discussion

The quantitative phase showed that two in five students have tried e-cigarettes, but occasional or daily use was reported by only 5% of them. Reasons for using e-cigarettes changed from experimentation to current use. While the experimentation was common and opportunistic, the current use was less frequent but rational. By focusing on e-cigarette users for at least two continuous months, the qualitative phase also showed that e-cigarettes were used by former smokers as a means either to switch to a new, lasting and pleasant behavior, or as a transitory step towards stopping smoking. Dual users, on the other hand, formed a heterogeneous group. They partly replaced tobacco use by vaping but with various perspectives, not necessarily to quit smoking.

Our prevalence estimates of e-cigarette use were close to those described in adult populations in other high-income countries with moderately restrictive e-cigarette policies [24]. A recent study suggests that binding regulations on e-cigarettes could influence use in the adult population [25]. However, it has not been established whether they also impact current use of e-cigarettes among young adults [26]. Interpretation of e-cigarette use levels should also take tobacco control policies into account. According to the 2014 Eurobarometer survey, e-cigarette experimentation and current use among subjects aged 15 or over were higher in European countries that have increased tobacco taxes or promoted aid for smoking cessation [27]. But the effectiveness of tobacco control policies was not similar in all high-income countries. While they have succeeded in reducing the smoking rates in Canada or Australia, it remains at a high level in other countries such as France or Romania. For policymakers, e-cigarettes could be seen as detrimental to efforts that have helped reduce smoking rates, except in countries where those remain high: they may be more tolerated there as harm reduction tools [28].

Our results suggested that becoming a current user was a choice supported by many reasons identified by students. Similar results have been described in a qualitative study led among Hawaiian young adults who were daily e-cigarette users. They reported the same variety of reasons for regular use of e-cigarettes: smoking cessation/reduction, health improvement, sensory satisfaction, self-regulation induced by nicotine psychotropic effects, convenience of indoor smoking, cleaner alternative to cigarette smoking, discreet use, recreational use, social enhancement, etc. [29]. The role played by nicotine addiction in pursuit of vaping was not obscured by our results. They only underlined the multifactorial nature of the installation in this practice in young adult population, like other studies. Smoking, e-cigarette use and nicotine addiction were measured for four years in a longitudinal study among US adults aged 19–23 [30]. Among never smokers at baseline, e-cigarette use was not significantly associated with subsequent tobacco use, either directly or mediated by nicotine addiction. In contrast, tobacco use at baseline was associated with subsequent e-cigarette use in smokers, both directly and through nicotine addiction. The transition from smoking to e-cigarette use was therefore only partially mediated by nicotine addiction.

Our results showed three e-cigarette user profiles among students. Identical profiles were described in another study conducted among UK vapers aged 19 to 69 [31]. A similar regulatory framework in the United Kingdom and France concerning e-cigarettes (European Tobacco Products Directive) partly explained the convergence of results. Although several studies described profiles of e-cigarette users in various national contexts [31–36], few focused specifically on young adults [35,36]. One was a qualitative study of 20 Americans aged 21 to 27 in Massachusetts, a smoke-free state with a lot of restrictions on vaping [36]. Authors identified four e-cigarette user profiles according to personal and social purposes. This study also included vapers who had never smoked before, unlike ours.

Our analyses had some limitations. Tobacco and e-cigarette use were self-reported in the quantitative phase, with a risk of underestimating prevalence due to memorization bias or social desirability. The low response rate, the predominance of women or freshmen among participants suggested selection bias. The predominance in the sample of students whose parents had a high level of education and were the main source of income also suggested that most of participants had a favorable socio-economic level. It was not possible to weight the prevalence estimators on variables allowing to appreciate the socio-economic level of the students because this information was not available about the target population. Moreover, we did not take into account the regulatory framework on the use of tobacco or e-cigarettes in the country of origin of the international students included. In the qualitative phase, more than half of participants were healthcare students. Being future health professionals might have influenced their discourse in favor of smoking cessation benefits. Moreover, no information to assess the socio-economic level of students was collected in the qualitative phase. The profiles of e-cigarette users in our analysis might appear frozen, but some prospective observational quantitative studies suggest that vaping is a more dynamic process in smokers, even in young adults [37–39]. Where follow-up was long enough ($\geq$ 12 months), multiple trajectories were observed with transitions from smoker to former smoker, from smoker to vaper, and so on. Finally, our studies were conducted in just one French university, limiting extrapolation to all French students or to young adults from countries with different e-cigarette regulations. Despite these weaknesses, our analysis seems to be the first to explore the relationship between e-cigarette and tobacco use in depth in a French student population, with a moderately restrictive regulation regarding e-cigarettes. With the quantitative study, we were able to estimate the weighted prevalence of experimentation and current use. The calibration method was used to reduce the effect of potential self-selection bias related to the voluntary participation of students which lead to an over-representation of women and freshmen in the quantitative phase. The qualitative study allowed us to propose profiles focusing on the lived experience of e-cigarette use.

## Conclusions

E-cigarette experimentation was frequent in this French student population, especially among smokers and former smokers. Current use was only reported by 5% of students. It was more reported by former and current smokers. The mixed approach provided a better understanding of the gap between this high level of experimentation and the relatively low level of current use. Moving from opportunistic experimentation to current use of e-cigarettes required having identified several arguments supporting this decision. Three distinct groups of users were identified: smoker (or dual user), former smoker and vaper. We also found that the current e-cigarette use was rare among never smoker students. Their reasons for continuing to use cigarettes and their identity characteristics were not explored in our qualitative study. They could deserve to be specifically studied in the "moderate" French regulatory context. Longitudinal quantitative or qualitative studies over more than one year in young adult smokers also appear

necessary to understand the dynamics of their tobacco and e-cigarette use, in particular changes in identity.

## Supporting information

**S1 Table. Vaping among college students who had tried e-cigarettes in a quantitative phase[α] (N = 704) and qualitative phase[β] (N = 20) from a mixed methods study carried out at the University of Bordeaux (France), 2016–2017.** *IQR: Interquartile range; [α]Data collected in the online quantitative study on e-cigarette use; [β]Data collected in qualitative research among former and current smokers who had used e-cigarettes for at least two continuous months.
(PDF)

**S1 File. Guide final en français.**
(PDF)

**S2 File. Final guide in English.**
(PDF)

**S3 File. Questionnaire d'inclusion dans le volet quantitatif en français.**
(PDF)

**S4 File. Baseline questionnaire in the quantitative phase in English.**
(PDF)

## Acknowledgments

The authors would like to thank Ms Jacqueline Pedley and Hancock Hutton Enterprise for copyediting the manuscript. They thank the "*GROupe Universitaire de recherche qualitative Médicale Francophone*" (GROUM.F) for training the first author in the concepts and techniques of qualitative research. They thank Chloé Leflot, Louis Bétolaud du Colombier, Lucile Roussel-Dupré and Cécile Landou who participated in the data collection and analysis in the qualitative research study. They acknowledge the research team of the i-Share project for their logistical support. Finally, they thank university services that have helped them with the research and are grateful to all the participants who volunteered to take part in the study.

## Author Contributions

**Conceptualization:** Shérazade Kinouani, Emmanuel Langlois, Christophe Tzourio.

**Data curation:** Shérazade Kinouani.

**Formal analysis:** Shérazade Kinouani, Héléna Da Cruz.

**Funding acquisition:** Christophe Tzourio.

**Investigation:** Shérazade Kinouani.

**Methodology:** Shérazade Kinouani, Emmanuel Langlois, Christophe Tzourio.

**Project administration:** Shérazade Kinouani.

**Software:** Shérazade Kinouani, Héléna Da Cruz.

**Supervision:** Shérazade Kinouani.

**Validation:** Shérazade Kinouani, Héléna Da Cruz, Emmanuel Langlois, Christophe Tzourio.

**Writing – original draft:** Shérazade Kinouani.

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
