## [Decision Letter · Decision Letter 0]

16 May 2022

PONE-D-21-33747Prevalence, lived experiences and user profiles in e-cigarette use: A mixed methods study among French college students.PLOS ONE

Dear Dr. KINOUANI,

Thank you for submitting your manuscript to PLOS ONE. After careful consideration, we feel that it has merit but does not fully meet PLOS ONE’s publication criteria as it currently stands. Therefore, we invite you to submit a revised version of the manuscript that addresses the points raised during the review process.

The manuscript topic is of high interest to scientific community, however there are some points raised by reviewers needs clarification and prompt response.  

We look forward to receiving your revised manuscript.

Kind regards,

Eman Sobh, M.D.

Academic Editor

PLOS ONE

Journal Requirements:

2. Please include additional information regarding all the survey, questionnaires or interview guides used in the study and ensure that you have provided sufficient details that others could replicate the analyses. For instance, if you developed a questionnaire as part of this study and it is not under a copyright more restrictive than CC-BY, please include a copy, in both the original language and English, as Supporting Information.

5. Thank you for stating the following financial disclosure: "This work was supported by the French National Cancer Institute [grant number INCa_11502]. The i-Share team is currently supported by an unrestricted grant of the Nouvelle-Aquitaine Regional Council (Conseil Régional Nouvelle-Aquitaine) [grant number: 4370420] and by the Bordeaux “Initiatives d’excellence” (IdEx) program of the University of Bordeaux [ANR-10-IDEX-03-02]. It has also received grants from the Nouvelle-Aquitaine Regional Health Agency (Agence Régionale de Santé Nouvelle-Aquitaine) and Public Health France (Santé Publique France). The funding bodies had no role in study design, data collection, analysis and interpretation, decision to publish or preparation of the manuscript."

We note that one or more of the authors is affiliated with the funding organization, indicating the funder may have had some role in the design, data collection, analysis or preparation of your manuscript for publication; in other words, the funder played an indirect role through the participation of the co-authors. If the funding organization did not play a role in the study design, data collection and analysis, decision to publish, or preparation of the manuscript and only provided financial support in the form of authors' salaries and/or research materials, please do the following:

a. Review your statements relating to the author contributions, and ensure you have specifically and accurately indicated the role(s) that these authors had in your study. These amendments should be made in the online form.

b. Confirm in your cover letter that you agree with the following statement, and we will change the online submission form on your behalf: 

“The funder provided support in the form of salaries for authors [insert relevant initials], but did not have any additional role in the study design, data collection and analysis, decision to publish, or preparation of the manuscript. The specific roles of these authors are articulated in the ‘author contributions’ section.

6. Please upload a new copy of Figure 1 as the detail is not clear. Please follow the link for more information: https://blogs.plos.org/plos/2019/06/looking-good-tips-for-creating-your-plos-figures-graphics/" https://blogs.plos.org/plos/2019/06/looking-good-tips-for-creating-your-plos-figures-graphics/

Reviewers' comments:

Reviewer's Responses to Questions

**Comments to the Author**

1. Is the manuscript technically sound, and do the data support the conclusions?

Reviewer #1: Partly

Reviewer #2: Yes

Reviewer #3: Yes

2. Has the statistical analysis been performed appropriately and rigorously? 

Reviewer #1: Yes

Reviewer #2: Yes

Reviewer #3: No

3. Have the authors made all data underlying the findings in their manuscript fully available?

Reviewer #1: No

Reviewer #2: Yes

Reviewer #3: Yes

4. Is the manuscript presented in an intelligible fashion and written in standard English?

Reviewer #1: Yes

Reviewer #2: Yes

Reviewer #3: Yes

5. Review Comments to the Author

Reviewer #1: The original sent for my review is a cross- sectional study aiming to describe the prevalence of e- cigarette users among French College students. This is an important topic because the use of these devices has led to significant debate on whether they should be considered as tobacco products or not and even if those devices are a safe way for smoking cessation.

However, due to study design, I cannot be sure that the authors have enrolled a representative population of French college students. If the authors selected those inscribed on a web page, this could have biased the population and therefore a true prevalence cannot be claimed by the authors.

Reviewer #2: The study is well-conceived, well-designed and well-performed. The conclusions are supported by results which in turn are supported by sound statistical methods. I have no major reservations or objections with the study.

Reviewer #3: An important topic had been studied. However, Topic guide if included would give a better idea of qualitative component. Thematic analysis is to be shown by reducing factors into major themes, as plan mentioned in methodology section but not seen in the results section. The socio- demographic factors can be analysed for their association with vaping.

6. PLOS authors have the option to publish the peer review history of their article (what does this mean?). If published, this will include your full peer review and any attached files.

Reviewer #1: **Yes: **Bernardino Alcazar- Navarrete

Reviewer #2: No

Reviewer #3: **Yes: **I have reviewed the manuscript entitled as "Prevalence, lived experiences and user profiles in e-cigarette use: A mixed methods study among French college students". I found this study a logically designed one but missing some important analysis and justification. The authors are advised to revise it according to the comments raised.

Dr, Farah Asad Mansuri

COnsultant Preventive Medicine

---

## [Author Response · Author response to Decision Letter 0]

14 Jul 2022

We revised the manuscript following the PLOS ONE's style requirements, including those for file naming. We used the Preflight Analysis and Conversion Engine (PACE) to improve our figures.

We have attached the baseline questionnaire of the i-Share project in French (S1 file) and in English lan-guage (S2 file). All questions about the tobacco and e-cigarette use asked as part of the ancillary study to the i-Share project are directly described in the manuscript (in the "quantitative component" section of Materials and methods).

We also added the final guide of our semi-structured interviews as supporting information, in French lan-guage (S3 file) and English language (S4 file).

We checked and corrected the "Funding information" section. We have also matched the two sections "Funding information" and "Financial disclosure". We also checked our statements relating to the author contributions, and indicated the roles that each author had in our study.

We apologize for the misunderstanding, but we only said that the interview guide was available on re-quest, not all the collected data. We have made the final guide available to readers as additional file 1 (in French language) and 2 (in English language). We have also provided the i-Share inclusion questionnaire as additional files 3 (in French language) and 4 (in English language).

Data cannot be shared freely as we have to comply to French regulations (the Commission Informatique et Libertés, which is the French authority that ensures that the collection of data in the context of re-search does not infringe on the freedoms, rights and privacy of individuals/number: DR-2013-019), and European regulations (General Data Protection Regulation). Thus, the qualitative component (inter-views) cannot be shared but data of the quantitative component of the study are available upon reason-able request to the authors. Our statement is therefore: The i-Share project data used in the quantitative component can be requested for scientific collaboration, following a procedure described on the study website (https://research.i-share.fr/how-to-collaborate/).

The quantitative component is indeed a cross-sectional study led in 2016 among French college students already participating in another study: the i-Share project. The i-Share project is an online study in which French-speaking students participate on a voluntary basis. This may result in a participation bias due to the self-selection of respondents. For this reason, we applied a method of calibration on margins to our data. This method allows the weight of each individual to be modified in our sample by taking into ac-count the real composition of the target population on auxiliary data (the calibration variables). Infor-mation on the real composition of the student population is provided by the French Ministry of Higher Education and Research. It is provided sparingly upon request by researchers.

In our analysis, the calibration variables were: gender, field of study and age in classes. This method al-lows for in-sample estimates after adjusting the margins of a contingency table to the margins of the target population on auxiliary data; these must be available for both the sample and the target popula-tion. The sample obtained after this weighting of individuals is then representative of the target popula-tion for the calibration variables.

We have added the reference of this calibration method to our revised manuscript : Deville JC, Särndal CE, Sautory O. Generalized raking procedures in survey sampling. J Am Stat Assoc. 1993;88(423):1013-20. 

To apply this statistic method, we used the MacroSAS Calmar® program. This program is described in detail in the "calibration methods" section by INSEE: https://www.insee.fr/en/information/5398341

The weighted estimators presented in Table 3 are therefore the closest estimators we can obtain of the true prevalence of e-cigarette use in the college student population of Bordeaux area, France.

We added our final guide as supporting information (S1 and S2 files). We have presented the categories related to the experience of vaping that emerged from the grounded theory analysis: becoming a current e-cigarette user; self-image; and social interactions. 

We modified the first part of the results to better highlight the main themes related to reasons for using e-cigarettes. We have added 2 figures to the manuscript to illustrate the organization into main themes on reasons for experimenting with e-cigarettes (figure 2) and reasons for current use (figure 3).

We could have described the profiles of regular e-cigarette users (those who have vaped for at least 2 months) by observing which socio-economic characteristics are associated with e-cigarette use. We did not choose this way of describing the profiles because it would not have allowed us to understand the diversity of students' experiences who don't give up after the first use. This is really what interested us because we think that this is both little studied and this diversity of experiences better explains the gap found between a strong experimentation but a weak current use in student population.

Introduction: 

We added some examples about smoking and vaping in the Introduction section. We also added information about the European young adults willing to quit smoking according to the Eurobarometer survey.

We finally added sentences before our objectives to explain why it seemed important in the context of French regulation of vaping to estimate the prevalence for using e-cigarettes and to study the reasons and practices of young adults to use them.

Materials and methods:

We used the chi square test to measure the association between smoking status and e-cigarette use (experimentation and current use). For this reason, we announced the application of this test in the method. We presented these measures of association in the text only so as not to overload Table 3. In addition, due to missing data on smoking status, the total sample for this stratified analysis is slightly different from that presented in Table 3 (N=1691 instead of 1694). Thus, we added a Table 4 in this revised manuscript with weighted prevalence of different use of e-cigarettes according to smoking status.

We apologize for this misunderstanding, but we never said that the sample for the qualitative study would be identical to that for the quantitative study. Our aim was to have a close-to-representative sample of the target population for the quantitative study and we applied a method of calibration on margins. However, for the qualitative study, what is sought is diversity of opinion, not representativeness. In grounded theory, it is recommended to vary the sample according to the hypotheses that emerge from the analysis of the interviews. This is why our sample was varied regarding the gender of the participants, their study fields and their smoking status. We thus followed the usual practice of “purposive sampling”. It did not emerge during our analysis that the age of the participants was a major factor explaining their vaping experience. We did not take this factor into account when diversifying our sample.

The quantitative study was led among students regardless of their smoking status and vaping status. The aim of this study was to estimate the prevalence of e-cigarette use and it would not have been relevant to exclude non-vapers. The qualitative study was led among regular e-cigarette users only (using at least 2 consecutive months). This choice can be explained by the desire to go deeper in the understanding of behaviors by describing the students' experience of vaping and the relationship they make between this use and their current or past use of tobacco. For this reason, we interviewed 7 students who had participated in the quantitative study, but we allowed ourselves to go beyond the framework of this participation. We increased our recruitment capacity by snowball strategy and by inviting students to participate also through social networks.

The interviews were rich: we wanted the students to discuss the reasons for using e-cigarettes but also their experience of vaping, from the first attempt to current use. The interviews lasted on average 55.25 minutes (minimum: 15 minutes; maximum: 120 minutes). We have added the average length of the interviews to the revised manuscript and the final guide in Supporting information.

Discussion:

We had already described the limitations of our work; we added a few sentences on the strengths.

Conclusion:

We first summarized the results of the quantitative and qualitative studies separately, and we added sentences to explain the importance or the mixed approach to understand why, among so many experimenters, few students remain in regular use.

We thank the reviewers and the editorial board for their comments and the opportunity to resubmit this manuscript after revision.

---

## [Decision Letter · Decision Letter 1]

26 Oct 2022

PONE-D-21-33747R1Prevalence, lived experiences and user profiles in e-cigarette use: a mixed methods study among French college students.PLOS ONE

Dear Dr. KINOUANI,

Thank you for submitting your manuscript to PLOS ONE. After careful consideration, we feel that it has merit but does not fully meet PLOS ONE’s publication criteria as it currently stands. Therefore, we invite you to submit a revised version of the manuscript that addresses the points raised during the review process.

One of the reviewers has suggested acceptance, indicating that all the comments and issues raised during the first round were satisfactorily addressed. However, the second reviewer has reservations and has invited further revisions before the manuscript can be accepted for publication. 

We look forward to receiving your revised manuscript.

Kind regards,

Lambros Lazuras

Academic Editor

PLOS ONE

Journal Requirements:

Reviewers' comments:

Reviewer's Responses to Questions

**Comments to the Author**

1. If the authors have adequately addressed your comments raised in a previous round of review and you feel that this manuscript is now acceptable for publication, you may indicate that here to bypass the “Comments to the Author” section, enter your conflict of interest statement in the “Confidential to Editor” section, and submit your "Accept" recommendation.

Reviewer #1: All comments have been addressed

Reviewer #3: (No Response)

2. Is the manuscript technically sound, and do the data support the conclusions?

Reviewer #1: Yes

Reviewer #3: Partly

3. Has the statistical analysis been performed appropriately and rigorously? 

Reviewer #1: Yes

Reviewer #3: No

4. Have the authors made all data underlying the findings in their manuscript fully available?

Reviewer #1: Yes

Reviewer #3: Yes

5. Is the manuscript presented in an intelligible fashion and written in standard English?

Reviewer #1: Yes

Reviewer #3: Yes

6. Review Comments to the Author

Reviewer #1: Authors have answered the questions made by this reviewer regarding selection bias and clarify their prior comments

Reviewer #3: Methodology and analysis for qualitative component of the study does not satisfy the scientific standards as such.

7. PLOS authors have the option to publish the peer review history of their article (what does this mean?). If published, this will include your full peer review and any attached files.

Reviewer #1: **Yes: **Bernardino Alcázar- Navarrete

Reviewer #3: **Yes: **Farah Mansuri

---

## [Author Response · Author response to Decision Letter 1]

31 Oct 2022

To:

Lambros Lazuras

Academic Editor

PLOS ONE

 Bordeaux, October 26th, 2022

Object: Response to reviewers’

Dear Pr Lambros Lazuras,

We are very grateful that you considered our article “Prevalence, lived experiences and user profiles in e-cigarette use: a mixed methods study among French college students” for revision despite the overall negative comment of reviewer #3.

We would have loved to respond point by point to reviewer #3 but this is made difficult because of the very short commentary provided by this reviewer.

Reviewer #3 answered “no” to question 3 about the appropriateness of the statistical analysis but the only comment given by the reviewer was that “the methodology and analysis for qualitative component of the study did not satisfy the scientific standards as such.” We will therefore concentrate on that, and we provide below explanations on the qualitative component and the final integrative phase.

1. We conducted a sequential explanatory mixed methods study (QUANT → qual). This study design described by Creswell JW et al (1,2) consists of collecting and analyzing quantitative and qualitative data separately and then interpreting them together. During this integrative phase, the qualitative results deepen one or more results obtained in the quantitative phase. As written by Ivankova NV et al. "the qualitative data are collected and analyzed second in the sequence and help explain, or elaborate on, the quantitative results obtained in the first phase" (2). 

This study design allows to deepen some results of the quantitative stage but its main weakness is the possible duration of the qualitative stage (1,2) as it is designed to answer questions emerging from the quantitative stage. This explains why the qualitative phase could take some time (which was noted by reviewer #3 during the initial evaluation). 

At the end of our quantitative stage, one question remained: as many student smokers try e-cigarettes, why few of them engage in its regular use? The quantitative phase, although taking into account various factors (socio-demographic, academic, use of psychoactive products, etc.) could not allow us to understand in depth why and how some students engage in regular vaping. We therefore clearly prioritize this question in our qualitative phase. As noted by Ivankova NV et al. : "depending on the study goals, the scope of quantitative and qualitative research questions, and the particular design of each phase, a researcher may give the priority to the qualitative data collection and analysis” (2). 

There are several interpretative approaches described in mixed-method studies for integrative phase (3). The connecting approach would impose a complete recruitment of the qualitative component participants among the participants of the quantitative component. The building approach that we chose requires that results from one data collection supports the next data collection. A partial connection of the two samples is possible if it makes it possible to deepen the results obtained during the first step.

2. Regarding the qualitative phase itself. We conducted two separate analyses of the twenty interviews. The first and main analysis was inspired by Grounded theory (4,5). Its objective was to answer to the question emerging at the end of the quantitative phase. We applied its main principles: openness, analyzing immediately, coding and the constant comparative method, theoretical sampling, theoretical saturation, and production of a substantive theory (5). 

We also conducted in parallel an inductive thematic analysis of the same interviews from a constructionist perspective. This supplementary analysis made by a group of three medical students (cited in the acknowledgments) supervised by the first author, allowed us to explore a secondary objective: to describe the reasons for experimenting with or continuing to use e-cigarettes. This supplementary analysis was possible because “thematic analysis is not wedded to any pre-existing theoretical framework, and therefore it can be used within different theoretical frameworks” said Braun V and Clarke V (6). The constructionist perspective assumes that “meaning and experience are socially produced and reproduced, rather than inhering within individuals” (6). As it was a secondary analysis, we have not detailed the data analysis procedure, so as not to lengthen the article. It was nevertheless carried out according to the six steps described by Braun and Clarke and resulted in figures 2 and 3. This double analysis of the interviews also explains why it was sometimes necessary to be 5 coders, as well as the duration perceived as unusually long for the qualitative component.

3. Participants selection. Seven volunteer students from the quantitative phase agreed to participate in the qualitative phase. However, it appeared during the analysis of these interviews that they alone did not make it possible to develop a sufficiently stable formal theory on how and why certain students commit themselves to vaping in France. As Ivankova NV et al. said "In the sequential explanatory design, a researcher typically connects the two phases while selecting the participants for the qualitative follow-up analysis based on the quantitative results from the first phase" (2). The selection of participants for the qualitative component must therefore consider the results of the quantitative component but it is not required that all participants in the qualitative study have participated in the other phase. What matters is that they are recruited according to characteristics that allow to answer the questions posed by the first stage. The differences between our two samples surprised the third reviewer in her first critical reading who pointed “qualitative study sample is said to be from quantitative one but differ a lot in its composition according to age and gender; that might affect interpretation of findings”. Our qualitative sample varied regarding the gender of the participants, their study fields, and their smoking status, following the usual practice of theoretical sampling. It did not emerge during our analysis that the age of the participants was a major factor explaining their vaping experience. We did not take this factor into account when diversifying our sample. Theorical sampling made possible to embed the two phases of our mixed-methods study by selecting participants based on the question posed by the first part and the ideas emerging during the analysis of the qualitative data.

4. Guidelines. We followed the COREQ guidelines for describing the qualitative component in the manuscript (7). 

5. Previous experience with qualitative studies. Our research team has already published few articles of qualitative research, in particular with mixed method research:

• Chamon Q Govindin Ramassamy K, Rahis AC, Guignot L, Tzourio C, Montagni I. Persistence of Vaccine Hesitancy and Acceptance of the EU Covid Certificate Among French Students. J Community Health 2022;47(4):666-73. doi: 10.1007/s10900-022-01092-6.

• Montagni I, Abraham M, Tzourio C, Luquiens A, Nguyen-Thanh V, Quatremere G. Mixed-methods evaluation of a prevention campaign on binge drinking and cannabis use addressed to young people. Journal of Substance Use 2022. doi: 10.1080/14659891.2021.2022223.

• Montagni I, Vialemaringe M, Tzourio C. Sport practice and perceptions in university students: a mixed-methods study. International Sport Science Student Studies 2020;2(1):1-15.

• Montagni I, Langlois E, Koman J, Petropoulos M, Tzourio C. Avoidance and Delay of Medical Care in the Young: An Interdisciplinary Mixed-methods Study. YOUNG 2017; 26(5):1-20. doi: 10.1177/1103308817734474.

One of our co-authors - Emmanuel Langlois - is a sociologist working on addictions and drug abuse in the young. He is a lecturer at the University of Bordeaux and his personal references can be found here: https://orcid.org/0000-0001-9733-5808

We hope that these explanations will assure you that we have conducted this study in the most scientifically rigorous manner possible. We also maintain that our conclusions are supported by the data. We believe that the misunderstanding of reviewer #3 on our qualitative analysis may be due to the lack of precision in the article about the two separate analyses led on the same qualitative data and we have added this precision in our revised version. The changes only affect the main text. Tables, figures, additional files, funding sources, author contributions and affiliations, acknowledgments, conflict of interest statement remain unaffected.

The work has not been published elsewhere, nor is it currently under consideration for publication elsewhere. All authors have sufficiently participated in the paper to fully support it and have approved this revised version. The authors declare that they have no conflicts of interest.

Thank you for considering our revised manuscript.

Yours sincerely,

Shérazade KINOUANI, Corresponding author

Université de Bordeaux, INSERM U1219, BPH, équipe HEALTHY

146, rue Léo Saignat - CS 61292 - 33076 Bordeaux Cedex France

sherazade.kinouani@u-bordeaux.fr

Phone: + (33)6.10.29.68.47

References

1. Creswell JW, Clark VLP. Designing and conducting mixed methods research. Thousand Oaks, CA, US: Sage Publications, Inc; 2007. 

2. Ivankova NV, Creswell JW, Stick SL. Using Mixed-Methods Sequential Explanatory Design: From Theory to Practice. Field Methods. 2006;18(1):3–20. 

3. Fetters MD, Curry LA, Creswell JW. Achieving Integration in Mixed Methods Designs—Principles and Practices. Health Serv Res. 2013;48(6pt2):2134–56. 

4. Glaser BG, Strauss AL. The discovery of grounded theory: strategies for qualitative research. 5th edition. New Brunswick, New Jersey: Aldine Transaction; 2010.

5. Sbaraini A, Carter SM, Evans RW, Blinkhorn A. How to do a grounded theory study: a worked example of a study of dental practices. BMC Med Res Methodol. 2011;11(1):128. 

6. Braun V, Clarke V. Using thematic analysis in psychology. Qual Res Psychol. 2006;3(2):77–101. 

7. Tong A, Sainsbury P, Craig J. Consolidated criteria for reporting qualitative research (COREQ): a 32-item checklist for interviews and focus groups. Int J Qual Health Care. 2007;19(6):349–57.

---

## [Decision Letter · Decision Letter 2]

23 May 2023

PONE-D-21-33747R2Prevalence, lived experiences and user profiles in e-cigarette use: a mixed methods study among French college students.PLOS ONE

Dear Dr. Kinouani,

Thank you for submitting your manuscript to PLOS ONE. After careful consideration, we feel that it has merit but does not fully meet PLOS ONE’s publication criteria as it currently stands. Therefore, we invite you to submit a revised version of the manuscript that addresses the points raised during the review process.The submitted manuscript requires major revision Please submit your revised manuscript by 7th June 2023. If you will need more time than this to complete your revisions, please reply to this message or contact the journal office at plosone@plos.org. Please include the following items when submitting your revised manuscript:A rebuttal letter that responds to each point raised by the academic editor and reviewer(s). You should upload this letter as a separate file labeled 'Response to Reviewers'.A marked-up copy of your manuscript that highlights changes made to the original version. You should upload this as a separate file labeled 'Revised Manuscript with Track Changes'.An unmarked version of your revised paper without tracked changes. You should upload this as a separate file labeled 'Manuscript'.

We look forward to receiving your revised manuscript.

Kind regards,

Lila Bahadur Basnet, MD

Guest Editor

PLOS ONE

Dear Authors,

The manuscript has been throughly reviewed by reviewers. Please find the feedbacks from the reviewers and make the necessary changes.

Thank you for the submission.

Best Wishes,

Lila Bahadur Basnet

Reviewers' comments:

Reviewer's Responses to Questions

**Comments to the Author**

1. If the authors have adequately addressed your comments raised in a previous round of review and you feel that this manuscript is now acceptable for publication, you may indicate that here to bypass the “Comments to the Author” section, enter your conflict of interest statement in the “Confidential to Editor” section, and submit your "Accept" recommendation.

Reviewer #1: All comments have been addressed

Reviewer #3: All comments have been addressed

Reviewer #4: (No Response)

2. Is the manuscript technically sound, and do the data support the conclusions?

Reviewer #1: Yes

Reviewer #3: Partly

Reviewer #4: Partly

3. Has the statistical analysis been performed appropriately and rigorously? 

Reviewer #1: Yes

Reviewer #3: I Don't Know

Reviewer #4: No

4. Have the authors made all data underlying the findings in their manuscript fully available?

Reviewer #1: Yes

Reviewer #3: Yes

Reviewer #4: Yes

5. Is the manuscript presented in an intelligible fashion and written in standard English?

Reviewer #1: Yes

Reviewer #3: Yes

Reviewer #4: Yes

6. Review Comments to the Author

Reviewer #1: No further comments from this reviewer . The authors have addressed the methodology issues regarding data collection & analysis

Reviewer #3: The authors have made conscious effort to improve the manuscript. I like to guide my dear authors that the comments are basically raised to improve the quality of scientific material and never meant to be negative or discouraging towards researchers.

But one thing is to be emphasized that qualitative studies are carried out in order to bring truth and a reality check in your findings. So, selection of participants very crucial and here, all the more you adopted a grounded theory approach that is very sensitive to type and number of participants in the area.

The methodological clarity has been explained a bit in this version.

Type of analysis mentioned by the authors is thematic analysis that is not adequately shown in results.

In fact, by reading the results, analysis fits more in narrative analysis rather content analysis or thematic analysis.

Results section of ABSTRACT still not incorporated the findings of qualitative component of study.

Conclusion as given in ABSTRACT also inappropriate as the objectives of qualitative component were not included in it.

Discussion should start with quantitative findings rather introspection of qualitative component.

Qualitative results were not adequately discussed

Reviewer #4: 1. The terms French Students/French Speaking Students used interchangeably here. This means French Speaking Student with French Nationality OR,

Also, include French Speaking International Students?

If the second is true, then the country specific Tobacco, e-cigarettes legislation should also be considered in exploring profiles of participants and their perception before making conclusion.

Also, respective sociodemographic variables should be taken into consideration which may have influence in behavioral outcomes. for e.g.: wealth quantile of students (as e-cigarettes are normally perceived as an "elite group's thing in developing countries). Poor families do not have access to e-cigarettes, at times.

2. Around 135 different questions are mentioned in baseline questionnaire tool, comprising of variables like: family background, current environment, technology, physical attributes, health, mental and sexual health, sleep, eating habits, use of psychoactive substances, alcohol use and medicines history. However, the outcomes of all these questions are not presented anywhere and the inter-relationships between such important variables are not shown. The importance of social determinants and globalization effects are highly underestimated in analysis. This reduces the quality of research as well as the trust among readers.

3. Cultural part (indeed a very important variable) mentioned in sub-topic of baseline questionnaire tool but no contents available anywhere in this regard. Why? Culture is an important part of human behavior. This cannot be ignored.

4. High no. of non-respondents noticed, i.e., from 5214 to 1698 respondents. Also, very few male respondents in Quantitative Section.

[So, a food for thought: Active male smokers and vapers may have intentionally avoided the survey which may have implications on prevalence too.]

Hence, exploring males’ perspective, specifically through qualitative components, is a must here, which is missing. Similarly, female respondents are higher in this quantitative survey. Did the study sequentially explore reasons through its qualitative components?

5. Low participation of senior students noticed:

This could be due to study pressure/stress after promotion to upper class. This cohort may have been more engaged in smoking / vaping but ignored the questionnaire/surveys. So, the results could apparently reflect “a tip of the iceberg”. More senior-male-students might be active smokers as well as using e-cigarettes in this study. Generally, junior students respond more frequently to such surveys than the senior students. New junior students may be at fear of unexpected disciplinary measures that may come along.

6. Gender wise splitting of all key variables is important. Also, age-group selection in both qualitative and quantitative study, should be same for the comparative analysis.

7. After having an option “beyond 3 years”, is it meaningful to also give “Others” as an option in academic year of study section ?

Methodology and Analysis:

8. The term in-depth interviews are more common in qualitative research to explore topics. In-depth interviews can be semi -structured or open. Open in-depth interviews are more important to dig in and explore various dimensions of “an important issue”, while semi-structured interviews guide the focus and explore some ideas of all topics in a controlled way. Some specific topics demand open-in-depth interviews. So, use of both as per context would be better to draw proper conclusions.

9. Use of Software/ Manual Thematic analysis can be optional to one another. Also, if using Nvivo software, updated versions are available with many qualitative analysis features. Such analysis is missing in this paper.

10. Analysis if done manually, must have themes and if done by Nvivo too, there must be thematic nodes. Better we mention these key themes/nodes serially and present findings and discussion in a systematic fashion under each themes/node. These themes/nodes, thus generated, are more important and can be compared with quantitative information obtained earlier, for triangulation and to identify deviation as well as direction of the content flow.

11. On top of manual thematic analysis and analysis by Nvivo thus, producing key nodes/themes, what is the purpose behind inductive thematic analysis here? (mere use of such words sounds good and looks fancy to readers but sometimes it can also equally counter us if we could not explain the procedure well. ["Inductive" = how can we say our analysis inductive? explain..]

12. Details on the process that we follow, the purpose and the contribution of any method matters more than mere numbers of popular methods used during analysis to show apparently rich analysis. Better to explain more here on what we did in inductive analysis... what are its added contribution to paper on top of manual/ thematic analysis conducted earlier and steps followed, to make it inductive...?? This should be visible in some way to readers. Details, if given would assure more trust.

13. Readers are often more interested to know the following on mixed methods: how the themes are generated during qualitative analysis, how are those themes sequentially linked to quantitative information produced beforehand, how the interviews reached saturation and how the conclusions are drawn. The methodological part of qualitative analysis is very important. As qualitative research on one hand, is considered more effective in exploring themes whereas on the other, the chances of deviation from the topic/focus also criticized by many. So, in addition to the response to above questions, limitations to be elaborated more, too.

14. It was mentioned by the authors that, Qualitative section considered Quantitative factors. But such factors that authors considered are not explicitly noticed in qualitative section. In contrast, important variables are missing in analysis.

15. Also, author mentioned analysis on reasons/e-cigarettes continuation as a secondary analysis. It is hard to agree on this statement.

16. Moreover, conclusion not much supported by the presented analysis.

To summarize, although author tried to see the status of French students in regard to e-cigarette experimentation and smoking behavior, this paper demands extensive analysis of inter-variable relationships to draw proper conclusion and rigorous methodological details before publication.

7. PLOS authors have the option to publish the peer review history of their article (what does this mean?). If published, this will include your full peer review and any attached files.

Reviewer #1: **Yes: **Bernardino Alcazar Navarrete

Reviewer #3: No

Reviewer #4: No

---

## [Author Response · Author response to Decision Letter 2]

13 Jul 2023

1. If the authors have adequately addressed your comments raised in a previous round of review and you feel that this manuscript is now acceptable for publication, you may indicate that here to bypass the “Comments to the Author” section, enter your conflict of interest statement in the “Confidential to Editor” section, and submit your "Accept" recommendation.

Reviewer #1: All comments have been addressed

Reviewer #3: All comments have been addressed

Reviewer #4: (No Response)

- We thank the reviewers for these comments.

2. Is the manuscript technically sound, and do the data support the conclusions? The manuscript must describe a technically sound piece of scientific research with data that supports the conclusions. Experiments must have been conducted rigorously, with appropriate controls, replication, and sample sizes. The conclusions must be drawn appropriately based on the data presented.

Reviewer #1: Yes

Reviewer #3: Partly

Reviewer #4: Partly

-We have rewritten the conclusion so that it is more in line with the objectives and results of the study, in particular those of the qualitative component.

3. Has the statistical analysis been performed appropriately and rigorously?

Reviewer #1: Yes

Reviewer #3: I Don't Know

Reviewer #4: No

-We hypothesize that the hesitation of reviewers 3 and 4, is related to the fact that we did not take economic variables into account to weight our prevalence estimators while the access to e-cigarettes is influenced by the economic level especially in youth and young adults (point 6 below). 

To weight our estimators, we applied a method of calibration on margins which allows the weight of each participant to be modified in our sample by taking into account the real composition of the target population on auxiliary data (the calibration variables) and the non-response bias. This method necessitates to have calibration variables available for both the sample and the target population. 

The calibration variables we used were: gender, field of study, and age in classes as they were both available in our sample and in the target population obtained through administrative databases. Unfortunately, the socio-economic level was not available in the target population and although we have this information in our sample, we couldn’t use it in our calibration method. 

We have added a comment on this limitation in our Discussion section “The predominance in the sample of students whose parents had a high level of education and were the main source of income also suggested that most of participants had a favorable socio-economic level. It was not possible to weight the prevalence estimators on variables allowing to appreciate the socio-economic level of the students because this information was not available about the target population”.

4. Have the authors made all data underlying the findings in their manuscript fully available? The PLOS Data policy requires authors to make all data underlying the findings described in their manuscript fully available without restriction, with rare exception (please refer to the Data Availability Statement in the manuscript PDF file). The data should be provided as part of the manuscript or its supporting information, or deposited to a public repository. For example, in addition to summary statistics, the data points behind means, medians and variance measures should be available. If there are restrictions on publicly sharing data—e.g. participant privacy or use of data from a third party—those must be specified.

Reviewer #1: Yes

Reviewer #3: Yes

Reviewer #4: Yes

-We thank the reviewers for these comments.

5. Is the manuscript presented in an intelligible fashion and written in standard English? PLOS ONE does not copyedit accepted manuscripts, so the language in submitted articles must be clear, correct, and unambiguous. Any typographical or grammatical errors should be corrected at revision, so please note any specific errors here.

Reviewer #1: Yes

Reviewer #3: Yes

Reviewer #4: Yes

-We thank the reviewers for these comments.

6. Review Comments to the Author

Reviewer #1: 

No further comments from this reviewer. The authors have addressed the methodology issues regarding data collection & analysis.

-We thank the reviewer for this comment.

Reviewer#2:

I am thankful to the editor to keep me in loop as a reviewer on this manuscript. The authors have made conscious effort to improve the manuscript. I like to guide my dear authors that the comments are basically raised to improve the quality of scientific material and never meant to be negative or discouraging towards researchers.

But one thing is to be ensured that qualitative studies are carried out in order to bring truth and a reality check in your findings. So, selection of participants very crucial as all the more you adopted a grounded theory approach that is very sensitive to type and number of participants.

The methodological clarity has been explained a bit in this version. 

Type of analysis mentioned by the authors is thematic analysis that is not adequately shown in results.

In fact, by reading the results, analysis fits more in narrative analysis rather content analysis or thematic analysis.

Results section still not incorporated the findings of qualitative component of study.

Conclusion also inappropriate as the objectives of qualitative component were not included in it.

DECSISION: MAJOR REVISION

-We thank the reviewer for this comment. His various comments have so far helped us considerably to improve the quality of the manuscript. We have modified the Methods section according the reviewer’s comments, especially for the qualitative phase. We have also chosen to present the Results section completely differently: we now present separately the results of the statistical analysis of the quantitative data, those of the thematic analysis (with its themes) and then, those of the Grounded Theory (with its conceptualizing categories). We ended with a paragraph dedicated to the integrative phase, combining the 3 analyses. We believe that this presentation incorporates and better highlights the results of the qualitative phase.

We hope that this new presentation removes the ambiguity on the analysis inspired by the Grounded Theory that we carried out (and not a narrative analysis or a narrative inquiry).

Finally, we corrected the Conclusions section and hope that it better incorporates the results of the qualitative phase.

Reviewer #3: 

The authors have made conscious effort to improve the manuscript. I like to guide my dear authors that the comments are basically raised to improve the quality of scientific material and never meant to be negative or discouraging towards researchers. But one thing is to be emphasized that qualitative studies are carried out in order to bring truth and a reality check in your findings. So, selection of participants very crucial and here, all the more you adopted a grounded theory approach that is very sensitive to type and number of participants in the area. The methodological clarity has been explained a bit in this version. Type of analysis mentioned by the authors is thematic analysis that is not adequately shown in results. In fact, by reading the results, analysis fits more in narrative analysis rather content analysis or thematic analysis. Results section of ABSTRACT still not incorporated the findings of qualitative component of study. Conclusion as given in ABSTRACT also inappropriate as the objectives of qualitative component were not included in it. Discussion should start with quantitative findings rather introspection of qualitative component. Qualitative results were not adequately discussed.

-We thank the reviewer for this comment. We have modified the Methods section according the reviewer’s comments, especially for the qualitative phase. We have also chosen to present the Results section completely differently: we now present separately the results of the statistical analysis of the quantitative data, those of the thematic analysis (with its themes) and then, those of the Grounded Theory (with its conceptualizing categories). We ended with a paragraph dedicated to the integrative phase, combining the 3 analyses. We believe that this presentation incorporates and better highlights the results of the qualitative phase.

We hope that this new presentation removes the ambiguity on the analysis inspired by the Grounded Theory that we carried out (and not a narrative analysis or a narrative inquiry).

We also corrected the abstract to better incorporates the results of the qualitative phase.

The Discussion section of the previous version of the article already started by presenting the results of the quantitative part. The paragraph about "Main findings" already provided weighted prevalence of e-cigarette use. The paragraph of "Comparison of results with the literature" already discussed the similarity of the estimators found with those described in other countries with high income levels. We also mentioned the need to interpret them, also considering the national context of regulation of tobacco and e-cigarette use.

We completed the Discussion section with a critique of the results of the qualitative phase, targeting the reasons for the current use of e-cigarettes and the user profiles.

Reviewer #4: 

6.1. The terms French Students/French Speaking Students used interchangeably here. This means French Speaking Student with French Nationality OR, Also, include French Speaking International Students?�If the second is true, then the country specific Tobacco, e-cigarettes legislation should also be considered in exploring profiles of participants and their perception before making conclusion.�Also, respective sociodemographic variables should be taken into consideration which may have influence in behavioral outcomes. for e.g.: wealth quantile of students (as e-cigarettes are normally perceived as an "elite group's thing in developing countries). Poor families do not have access to e-cigarettes, at times.

-We thank the reviewer for this comment. “French students” included in our study both French Speaking Student with French Nationality and French Speaking International Students. Indeed, any student who understood French could participate in the i-Share Project. However, we restricted the current study to students living in France and enrolled at the University of Bordeaux . The legislation on tobacco and e-cigarettes was therefore the same for all the students included in our analyses.

We share the opinion of the reviewer that "e-cigarettes are normally perceived as an "elite group's thing in developing countries and poor families do not have access to e-cigarettes, at times".

We had little simultaneous information on our sample and the target population to apply a method of calibration on margins. As explained above, we had no information on students’ socio-economic level on the target population (administrative sample). We added sentences about this limit in section Discussion” It was not possible to weight the prevalence estimators on variables allowing to appreciate the socio-economic level of the students because this information was not available about the target population”.

6.2. Around 135 different questions are mentioned in baseline questionnaire tool, comprising of variables like: family background, current environment, technology, physical attributes, health, mental and sexual health, sleep, eating habits, use of psychoactive substances, alcohol use and medicines history. However, the outcomes of all these questions are not presented anywhere and the inter-relationships between such important variables are not shown. The importance of social determinants and globalization effects are highly underestimated in analysis. This reduces the quality of research as well as the trust among readers.

-As in all cohort studies, there are many variables collected in the i-Share project. However, this rich collection has various purposes. The cohort is indeed a collaboration platform for different research teams, studying different thematic fields. For our quantitative component, we selected the variables available at the time of data extraction and which seemed to us to be the most relevant because they had already been identified in the literature on e-cigarettes. Our goal was not to test all the links between these variables and the use of e-cigarettes without a priori assumptions. This would use other methods, for example machine learning as we have done in other articles from i-Share data: Macalli M, Navarro M, Orri M, Tournier M, Thiébaut R, Côté SM, Tzourio C. A machine learning approach for predicting suicidal thoughts and behaviours among college students. Sci Rep. 2021;11(1):11363. doi: 10.1038/s41598-021-90728-z.

We agree with the reviewer: there may be hidden or complex relationships (interactions) between certain variables, between them and with e-cigarette use. However, the models we used and the hypothesis-driven method involved limiting the number of potential risk factors. On the one hand, we were exposing ourselves to an inflation of the Alpha risk; on the other hand, we ran the risk of incidental findings of significant associations due to chance.

We also completely agree with the reviewer on the importance of the social determinants which we have tried to take into account as well as possible in our study, with the limits that we have exposed above. We have added sentences in the Discussion section on our difficulty of taking these social factors into account in an exhaustive way; this is a limitation of our conclusions.

6.3. Cultural part (indeed a very important variable) mentioned in sub-topic of baseline questionnaire tool but no contents available anywhere in this regard. Why? Culture is an important part of human behavior. This cannot be ignored.

-We apologize for this misunderstanding. In the inclusion questionnaire, question n°70 (in S4 file) dealing with cultural activities is a question about students' extracurricular "cultural" activities: theater, cinema, music, etc. It is neither a question on ethnic origins (the collection of this information is prohibited in research in France), nor a question on membership of a particular behavioral community group: LGBTQIA+, religious community, etc. This type of information was not collected in the i-Share project.

6.4. High no. of non-respondents noticed, i.e., from 5214 to 1698 respondents. Also, very few male respondents in Quantitative Section.[So, a food for thought: Active male smokers and vapers may have intentionally avoided the survey which may have implications on prevalence too.]Hence, exploring males’ perspective, specifically through qualitative components, is a must here, which is missing. Similarly, female respondents are higher in this quantitative survey. Did the study sequentially explore reasons through its qualitative components?

-We were aware of this greater participation of women and very young students in the i-Share project. It is also for this reason that we chose gender and age (in classes) as calibration variables in the quantitative study. We have also considered the low response rate of 34% by choosing as the margin calibration program the one developed by INSEE, which precisely takes into account the non-response bias: the MacroSAS Calmar® program.

In the qualitative part, we took care to involve as many women as men by diversifying the theoretical purposive sample on gender throughout the study (table 4). We also interviewed as many students aged 18-25 as students aged 26-29 for the same reason. 

The same work of diversification has been done regarding the field of study and smoking status. This assumed selection of the sample for the qualitative component allowed us to tend - for this time - not towards representativeness but the diversity of the opinions collected.

6.5. Low participation of senior students noticed: This could be due to study pressure/stress after promotion to upper class. This cohort may have been more engaged in smoking / vaping but ignored the questionnaire/surveys. So, the results could apparently reflect “a tip of the iceberg”. More senior-male-students might be active smokers as well as using e-cigarettes in this study. Generally, junior students respond more frequently to such surveys than the senior students. New junior students may be at fear of unexpected disciplinary measures that may come along.

-We agree with the reviewer's comment. That is why the calibration method on margins precisely took into account both age and gender, assigning individual weights to each participant in our sample according to what is known in the target population about the calibration variables.

6.6. Gender wise splitting of all key variables is important. Also, age-group selection in both qualitative and quantitative study, should be same for the comparative analysis.

-We agree with the reviewer's comment. Age and gender are major explanatory factors for the use of tobacco and e-cigarettes. It was essential to take this into account in each part of the study:

- Our sample of the quantitative component was initially not representative of students from the University of Bordeaux in terms of age, gender and field of study. This was a problem for estimating prevalence. We applied the method of calibration on margins on our sample considering these 3 variables to tend towards representativeness before calculating the expected estimators.

- The objective in the qualitative section was not to collect opinions representative of the target population (University of Bordeaux). This time, the priority was to collect the diversity of opinions. We knew that more women and younger students had participated in the quantitative component. The research team decided to give the floor to both women and men in the qualitative component, as much to junior students as to older ones. Qualitative methods (in particular the Grounded Theory method) precisely made it possible to voluntarily select the sample according to the subject treated and the hypotheses that emerge as the semi-structured interviews were analyzed.

Thus, the way in which we took these two factors into account differed between the 2 components and explained the non-comparable composition of the 2 samples. While we have sought to reduce the over-representation of women and young students in the quantitative section by applying the margin calibration method, we have endeavored to have the most diversified sample in terms of age and gender.

6.7. After having an option “beyond 3 years”, is it meaningful to also give “Others” as an option in academic year of study section?

-When completing the inclusion questionnaire, some students did not know how to answer about their academic year of study. We offered them to provide information in free text and the research team then took care of recoding it. For 28 students retained in the quantitative phase, we were unable to recode the information in free text. We wore them in "Other".

6.8. The term in-depth interviews are more common in qualitative research to explore topics. In-depth interviews can be semi -structured or open. Open in-depth interviews are more important to dig in and explore various dimensions of “an important issue”, while semi-structured interviews guide the focus and explore some ideas of all topics in a controlled way. Some specific topics demand open-in-depth interviews. So, use of both as per context would be better to draw proper conclusions.

-We agree with the reviewer's opinion on the value of conducting in-depth open interviews. However, the students of the research team who led interviews were trained in qualitative methods but had little experience in conducting open-ended interviews. It was important for us that these students led the interviews because of their proximity in age to the respondents (unlike the rest of the research team). Thus, we only performed semi-structured interview, collection method most mastered by our students in charge of data collection.

6.9. Use of Software/ Manual Thematic analysis can be optional to one another. Also, if using Nvivo software, updated versions are available with many qualitative analysis features. Such analysis is missing in this paper.

-We left the choice to the people of the research team on the use or not of an analysis software. The two students of the research team involved in the data analysis chose to use the NVivo°software. They were not asked to use the multiple functionalities of the software to complete the analyses carried out.

6.10. Analysis if done manually, must have themes and if done by Nvivo too, there must be thematic nodes. Better we mention these key themes/nodes serially and present findings and discussion in a systematic fashion under each themes/node. These themes/nodes, thus generated, are more important and can be compared with quantitative information obtained earlier, for triangulation and to identify deviation as well as direction of the content flow.

-We have totally changed the presentation in the Results section. We now present separately the results of the statistical analysis of the quantitative data, those of the thematic analysis and then, those of the Grounded Theory. We ended with a paragraph dedicated to the integrative phase. We have in this new presentation clearly announced the different themes of thematic analysis as well as the different conceptualizing categories of the Grounded Theory analysis.

6.11. On top of manual thematic analysis and analysis by Nvivo thus, producing key nodes/themes, what is the purpose behind inductive thematic analysis here? (mere use of such words sounds good and looks fancy to readers but sometimes it can also equally counter us if we could not explain the procedure well. ["Inductive" = how can we say our analysis inductive? explain..]

-We mean by the inductive approach the fact of making the themes emerged from the analysis of the data alone. We were not looking to see during the analysis if pre-established themes from a theoretical framework were found.

We formulated few initial hypotheses for the thematic analysis on the reasons for using e-cigarettes. The only assumptions made came from the analysis of the quantitative data. On the other hand, we had no hypothesis at the start of the analysis inspired by Grounded Theory about the categories that would emerge from the analysis of data targeting current use (lived experience and user profiles).

6.12. Details on the process that we follow, the purpose and the contribution of any method matters more than mere numbers of popular methods used during analysis to show apparently rich analysis. Better to explain more here on what we did in inductive analysis... what are its added contribution to paper on top of manual/ thematic analysis conducted earlier and steps followed, to make it inductive...?? This should be visible in some way to readers. Details, if given would assure more trust.

-We did not seek to unnecessarily multiply popular methods of analyzing qualitative data in this work. Thematic analysis was chosen to study the reasons for using e-cigarettes because we thought that this method would allow us to better highlight convergences and divergences between the reasons for experimenting with e-cigarettes and those for continuing to use them. Our thematic analysis showed that although the 3 main themes were similar for experimentation and current use, vapers' descriptions of these themes were richer when talking about their current use. This thematic analysis also confirmed the opportunistic nature of the experiment shown in the quantitative phase.

The analysis inspired by Grounded Theory seemed to us more appropriate to bring out the conceptualizing categories describing the lived experiences of vapers or their user profiles. This method makes it possible to generate conceptualizing categories, to create axes connecting them and to lead to a theorization about the current use of e-cigarettes.

We hope that with the new presentation of the results (especially the paragraph dedicated to the integrative analysis), the interest of the different analytical approaches will be more visible.

6.13. Readers are often more interested to know the following on mixed methods: how the themes are generated during qualitative analysis, how are those themes sequentially linked to quantitative information produced beforehand, how the interviews reached saturation and how the conclusions are drawn. The methodological part of qualitative analysis is very important. As qualitative research on one hand, is considered more effective in exploring themes whereas on the other, the chances of deviation from the topic/focus also criticized by many. So, in addition to the response to above questions, limitations to be elaborated more, too.

-We have provided some additional information in the Methods section:

- We have added references on thematic analysis and Grounded Theory: Braun V, Clarke V. Using thematic analysis in psychology. Qual Res Psychol. 2006;3(2):77–101. Corbin J, Strauss A. Basics of qualitative research: Techniques and procedures for developing grounded theory. Thousand Oaks, CA, US: Sage Publications, Inc; 2008. 

- We specified when the saturation was reached "At the end of the eighteenth interview, we seemed to have reached data saturation. Two other interviews were added, without any new themes or categories emerging";

- We specified in the paragraph on the integrative phase how the link was made between the quantitative component and the qualitative component;

We finally insist on the bridges between quantitative and qualitative aspects in the Results section (paragraph on the integrative phase).

6.14. It was mentioned by the authors that, Qualitative section considered Quantitative factors. But such factors that authors considered are not explicitly noticed in qualitative section. In contrast, important variables are missing in analysis.

-We thank the reviewer for his comments. We have rewritten the paragraph on the integrative phase in the Methods and Results sections, to provide clarification.

The reasons for using e-cigarettes (experimentation and current use) found in the quantitative part indeed were taken into account in the qualitative component. The students in charge of conducting the semi-structured interviews challenged the respondents on one or other of the reasons mentioned in the quantitative part. As shown in the S1 Table, figures 2 and 3, this made it possible to find during the interviews reasons for using the e-cigarette already identified but also to bring out new ones.

The sociodemographic, economic, academic, and medical characteristics of students in the quantitative component were not all used in the qualitative component. Only age, smoking status, gender and field of study were used to diversify the sample in the qualitative component. The diversification based on field of study had been difficult: we had many health students who wanted to participate to interviews. The collection of data in the qualitative study took longer than expected (more than 1 year) to succeed in interviewing students from other study fields.

6.15. Also, author mentioned analysis on reasons/e-cigarettes continuation as a secondary analysis. It is hard to agree on this statement.

-We apologize for the misunderstanding generated by presenting this additional thematic analysis. We have modified the paragraph of the Methods section, presenting the two analyses carried out on the qualitative data like performed separately but simultaneously.

6.16. Moreover, conclusion not much supported by the presented analysis. 

-We corrected the Conclusions section and hope that it better incorporates the results of all presented analyses.

To summarize, although author tried to see the status of French students in regard to e-cigarette experimentation and smoking behavior, this paper demands extensive analysis of inter-variable relationships to draw proper conclusion and rigorous methodological details before publication.

-We thank the reviewer for all these comments. We have tried to improve the presentation of the Methods and Results sections. We also have discussed the limitations identified by the reviewer in the Discussion section.

---

## [Decision Letter · Decision Letter 3]

30 Nov 2023

PONE-D-21-33747R3Prevalence, lived experiences and user profiles in e-cigarette use: a mixed methods study among French college students.PLOS ONE

Dear Dr. KINOUANI,

Thank you for submitting your manuscript to PLOS ONE. After careful consideration, we feel that it has merit but does not fully meet PLOS ONE’s publication criteria as it currently stands. Therefore, we invite you to submit a revised version of the manuscript that addresses the points raised during the review process.

  Please take note of the requested modifications from one of the two reviewers. While they are recommendations, it is helpful to either make those modifications or submit adequate justification for not making the requested changes.

We look forward to receiving your revised manuscript.

Kind regards,

Mark Allen Pershouse, PhD

Academic Editor

PLOS ONE

Journal Requirements:

Reviewers' comments:

Reviewer's Responses to Questions

**Comments to the Author**

1. If the authors have adequately addressed your comments raised in a previous round of review and you feel that this manuscript is now acceptable for publication, you may indicate that here to bypass the “Comments to the Author” section, enter your conflict of interest statement in the “Confidential to Editor” section, and submit your "Accept" recommendation.

Reviewer #1: All comments have been addressed

Reviewer #4: All comments have been addressed

2. Is the manuscript technically sound, and do the data support the conclusions?

Reviewer #1: Yes

Reviewer #4: Yes

3. Has the statistical analysis been performed appropriately and rigorously? 

Reviewer #1: Yes

Reviewer #4: Yes

4. Have the authors made all data underlying the findings in their manuscript fully available?

Reviewer #1: Yes

Reviewer #4: Yes

5. Is the manuscript presented in an intelligible fashion and written in standard English?

Reviewer #1: Yes

Reviewer #4: Yes

6. Review Comments to the Author

Reviewer #1: No further comments raised by this reviewer. The authors have improved the overall quality of their work

Reviewer #4: Well, Great! Much improved version.

Still, it would have been better if, country specific tobacco legislation has been reviewed for some significant outcomes (Desk review of some countries of participants that are significant would do). This would even give us some interesting insights/findings. Behavioral changes take time and duration of stay at France matters. So, merely students living in France and enrolled at the University of Bordeaux does not justify this. For e.g. A student may have just enrolled in the University and living in France just recently during this study and his/her habits largely depend on his previous environment and legal framework of hometown.

However, the justification made on the next point that the data shared from the i-Share Project have some limitations including that of socio-economic parameters is fine.

To conclude, all the justifications made in this version of paper, must come at least somewhere in the paper for the readers to understand the scenario better. Once this is done in addition to some minor changes, and the revision also in the abstract accordingly, I feel this manuscript could proceed for publication process. Thank you so much for clear explanation on each part which was missing initially. These minute details once added, would make methodological section strong.

Appreciated the meticulous work of the authors. I am also thankful to the editor for this reviewing opportunity.

7. PLOS authors have the option to publish the peer review history of their article (what does this mean?). If published, this will include your full peer review and any attached files.

Reviewer #1: **Yes: **Bernardino Alcázar Navarrete

Reviewer #4: No

---

## [Author Response · Author response to Decision Letter 3]

11 Dec 2023

Journal Requirements:

We have checked all references, and none are related to a retracted article.

Reviewers' comments:

Reviewer's Responses to Questions

Comments to the Author

1. If the authors have adequately addressed your comments raised in a previous round of review and you feel that this manuscript is now acceptable for publication, you may indicate that here to bypass the “Comments to the Author” section, enter your conflict of interest statement in the “Confidential to Editor” section, and submit your "Accept" recommendation.

Reviewer #1: All comments have been addressed

Reviewer #4: All comments have been addressed

We thank the reviewers for these comments.

2. Is the manuscript technically sound, and do the data support the conclusions?

Reviewer #1: Yes

Reviewer #4: Yes

We thank the reviewers for these comments.

3. Has the statistical analysis been performed appropriately and rigorously?

Reviewer #1: Yes

Reviewer #4: Yes

We thank the reviewers for these comments.

4. Have the authors made all data underlying the findings in their manuscript fully available?

Reviewer #1: Yes

Reviewer #4: Yes

We thank the reviewers for these comments.

5. Is the manuscript presented in an intelligible fashion and written in standard English?

Reviewer #1: Yes

Reviewer #4: Yes

We thank the reviewers for these comments.

6. Review Comments to the Author

Reviewer #1: No further comments raised by this reviewer. The authors have improved the overall quality of their work

We thank the reviewer for this comment.

Reviewer #4: Well, Great! Much improved version.

Still, it would have been better if, country specific tobacco legislation has been reviewed for some significant outcomes (Desk review of some countries of participants that are significant would do). This would even give us some interesting insights/findings. Behavioral changes take time and duration of stay at France matters. So, merely students living in France and enrolled at the University of Bordeaux does not justify this. For e.g. A student may have just enrolled in the University and living in France just recently during this study and his/her habits largely depend on his previous environment and legal framework of hometown.

However, the justification made on the next point that the data shared from the i-Share Project have some limitations including that of socio-economic parameters is fine.

To conclude, all the justifications made in this version of paper, must come at least somewhere in the paper for the readers to understand the scenario better. Once this is done in addition to some minor changes, and the revision also in the abstract accordingly, I feel this manuscript could proceed for publication process. Thank you so much for clear explanation on each part which was missing initially. These minute details once added, would make methodological section strong. 

Appreciated the meticulous work of the authors. I am also thankful to the editor for this reviewing opportunity.

We thank the reviewer for this comment. At the end of the 2nd revision, we already implemented all the responses to the reviewers to the manuscript: modification of the presentation of the Method or Result sections; more details on the methods applied in the quantitative and qualitative phases; addition of a paragraph on the integrative phase; enrichment of the Discussion section on the limits of the study.

We added in this 3rd revised version:

- A sentence specifying the experience of medical students in conducting interviews in the Methods section “Two trained medical students involved in the research team led all interviews because of their proximity in age to the respondents”.

- A sentence specifying that the program macroSAS CALMAR® applied in the quantitative phase considers the non-response bias in the Methods section “This calibration was carried out with a program developed by the French National Institute for Statistics and Economic Studies designed to take into account the non-response bias: the macroSAS CALMAR® [20]”.

- A paragraph in the Methods section about the benefit of having used two different analytical approaches in the qualitative phase “Thematic analysis is an analytical method allowing to both test the motives for using e-cigarettes identified in the quantitative phase and to highlight convergences and divergences between the motives for experimenting and those for continuing to use electronic cigarettes. On the other hand, the analysis inspired by Grounded Theory seemed more appropriate to bring out the conceptualizing categories describing the lived experiences of vapers or their user’s profile”.

- A sentence in the Discussion section underlining the interest of calibration on margins to reduce the potential effect of the over-representation of women and freshmen in the quantitative phase “The calibration method was used to reduce the effect of potential self-selection bias related to the voluntary participation of students which lead to an over-representation of women and freshmen in the quantitative phase”.

- A comment in the Discussion section on our lack of consideration of the regulatory framework of the origin country of international students included in our study “Moreover, we did not take into account the regulatory framework on the use of tobacco or e-cigarettes in the country of origin of the international students included “.

- Finally, we modified the abstract to indicate with one sentence the presence of limits in our study “Despite some limitations mainly related to the participants self-selection, this research showed that …”

---

## [Decision Letter · Decision Letter 4]

2 Jan 2024

Prevalence, lived experiences and user profiles in e-cigarette use: a mixed methods study among French college students.

PONE-D-21-33747R4

Dear Dr. KINOUANI,

We’re pleased to inform you that your manuscript has been judged scientifically suitable for publication and will be formally accepted for publication once it meets all outstanding technical requirements.

Kind regards,

Mark Allen Pershouse, PhD

Academic Editor

PLOS ONE

Additional Editor Comments (optional):

Reviewers' comments:

Reviewer's Responses to Questions

**Comments to the Author**

1. If the authors have adequately addressed your comments raised in a previous round of review and you feel that this manuscript is now acceptable for publication, you may indicate that here to bypass the “Comments to the Author” section, enter your conflict of interest statement in the “Confidential to Editor” section, and submit your "Accept" recommendation.

Reviewer #1: All comments have been addressed

Reviewer #4: All comments have been addressed

2. Is the manuscript technically sound, and do the data support the conclusions?

Reviewer #1: Yes

Reviewer #4: Yes

3. Has the statistical analysis been performed appropriately and rigorously? 

Reviewer #1: Yes

Reviewer #4: Yes

4. Have the authors made all data underlying the findings in their manuscript fully available?

Reviewer #1: No

Reviewer #4: Yes

5. Is the manuscript presented in an intelligible fashion and written in standard English?

Reviewer #1: Yes

Reviewer #4: Yes

6. Review Comments to the Author

Reviewer #1: Authors have answered the requested questions/modifications from this reviewer.

No further comments needed.

Reviewer #4: It is my pleasure to be a part of this review process. I am thankful to both the editor and the author's team for this opportunity. I think , the manuscipt can now proceed for the publication process.

7. PLOS authors have the option to publish the peer review history of their article (what does this mean?). If published, this will include your full peer review and any attached files.

Reviewer #1: **Yes: **Bernardino Alcázar- Navarrete

Reviewer #4: No

---

## [Editor Report · Acceptance letter]

1 Feb 2024

PONE-D-21-33747R4 

PLOS ONE

Dear Dr. Kinouani, 

I'm pleased to inform you that your manuscript has been deemed suitable for publication in PLOS ONE. Congratulations! Your manuscript is now being handed over to our production team.

Kind regards, 

on behalf of

Dr. Mark Allen Pershouse 

Academic Editor

PLOS ONE